

# Arctic Mediterranean Exchanges: A consistent volume budget and trends in transports from two decades of observations

Svein Østerhus[1], Rebecca Woodgate[2], Héðinn Valdimarsson[3], Bill Turrell[4], Laura de Steur[5], Detlef Quadfasel[6], Steffen M. Olsen[7], Martin Moritz[6], Craig M. Lee[2], Karin Margretha H. Larsen[8], Steingrímur Jónsson[9], Clare Johnson[10], Kerstin Jochumsen[6], Bogi Hansen[8], Beth Curry[2], Stuart Cunningham[10], Barbara Berx[4]

[1]NORCE Norwegian Research Centre, Bergen, Norway
[2]University of Washington, Seattle, USA
[3]Marine and Freshwater Research Institute, Reykjavik, Iceland
[4]Marine Scotland Science, Marine Laboratory, Aberdeen, UK
[5]Royal Netherlands Institute for Sea Research NIOZ, Texel, The Netherlands
[6]Institut für Meereskunde, Universität Hamburg, Germany
[7]Danish Meteorological Institute, Copenhagen, Denmark
[8]Faroe Marine Research Institute, Tórshavn, Faroe Islands
[9]Marine and Freshwater Research Institute & University of Akureyri, Iceland
[10]Scottish Association for Marine Science, Oban, UK

*Correspondence to*: Svein Østerhus (svein.osterhus@norceresearch.no)

**Abstract.** The Arctic Mediterranean (AM) is the collective name for the Arctic Ocean, the Nordic Seas, and their adjacent shelf seas. Into this region, water enters through the Bering Strait (Pacific inflow) and through the passages across the Greenland-Scotland Ridge (Atlantic inflow) and then modified within the AM. The modified waters leave the AM in several flow branches, which are grouped into two different categories: (1) overflow of dense water through the deep passages across the Greenland-Scotland Ridge, and (2) outflow of light water – here termed surface outflow – on both sides of Greenland. These exchanges transport heat, salt, and other substances into and out of the AM and are important for conditions in the AM. They are also part of the global ocean circulation and climate system. Attempts to quantify the transports by various methods have been made for many years, but only recently, has the observational coverage become sufficiently complete to allow an integrated assessment of the AM-exchanges based solely on observations. In this study, we focus on the transport of water and have collected data on volume transport for as many AM-exchange branches as possible between 1993-2015. The total AM-import (oceanic inflows plus freshwater) is found to be 9.1±0.7 Sv (1 Sv = 106 m3 s-1) and has a seasonal variation of amplitude close to 1 Sv and maximum import in October. Roughly one third of the imported water leaves the AM as surface outflow with the remaining two thirds leaving as overflow. The overflow is mainly produced from modified Atlantic inflow and around 70% of the total Atlantic inflow is converted into overflow, indicating a strong coupling between these two exchanges. The surface outflow is fed from the Pacific inflow and freshwater, but is still ~ 2/3rds from modified Atlantic water. For the inflow branches and the two main overflow branches (Denmark Strait and Faroe Bank Channel), systematic monitoring of volume transport has been established since the mid-1990s and this allows us to estimate trends for the AM-exchanges as a whole. At the 95 % level, only the inflow of Pacific water through the Bering Strait showed a statistically significant trend,





which was positive. Both the total AM-inflow and the combined transport of the two main overflow branches also showed trends consistent with strengthening, but they were not statistically significant. They do suggest, however, that any significant weakening of these flows during the last two decades is unlikely and the overall message is that the AM-exchanges remained remarkably stable in the period from the mid-1990s to the mid-2010s. The overflows are the densest source water for the deep

limb of the North Atlantic part of the Meridional Overturning Circulation (AMOC), and this conclusion argues that the reported weakening of the AMOC was not due to overflow weakening or reduced overturning in the AM. Although the combined data set has made it possible to establish a consistent budget for the AM-exchanges, the observational coverage for some of the branches is limited, which introduces considerable uncertainty. This lack of coverage is especially extreme for the surface outflows through the Denmark Strait, the overflow across the Iceland-Faroe Ridge, and the inflow over the Scottish shelf. We

recommend that more effort is put into observing these flows as well as maintaining the monitoring systems established for the other exchange branches.

# 1 Introduction

In most directions, the Arctic Mediterranean (AM) is surrounded by landmasses - Eurasia, North America, and Greenland - but a number of gaps connect the AM to the rest of the World Ocean. The connection to the Pacific is the Bering Strait,

while connections to the Atlantic[1] are through the Canadian Archipelago and through the gaps between Greenland and the European continent (Figure 1). Through these gaps, flows pass into and out of the AM, transporting water, heat, salt, and other substances. Here, our focus is only on the transport of water (volume), not e.g., heat or freshwater fluxes. The main aim of this manuscript is to synthesize the available observational evidence of the volume transports of these flows and their variability into a consistent budget and then to identify possible trends.

Though heat exchanges are the focus of regional climate studies, AM-exchanges also play an important role in the global climate through their influence on the Atlantic Meridional Overturning Circulation (AMOC). Between Greenland and the European continent, warm saline water flows from the Atlantic into the AM where it is cooled via air-sea exchange processes. The waters are also freshened by runoff, net precipitation and mixing with Pacific waters (and ice melt), but still much of the resulting water mass is sufficiently dense to be transported to greater depths through various processes

(e.g. Rudels et al., 1999).

These dense water masses leave the AM through the deep passages across the Greenland-Scotland Ridge and enter the Atlantic as "overflow" waters. They are much denser than the ambient water masses in the Atlantic and descend to deeper levels to form bottom intensified boundary currents. Together with the ambient waters from the Atlantic that they entrain on route, the overflow waters are understood to contribute the main component of the North Atlantic Deep Water

(NADW), (Gebbie and Huybers, 2010) which constitutes the deep branch of the AMOC (Dickson and Brown, 1994; Hansen

---

[1] Strictly speaking, all of the AM is part of the Atlantic Ocean, but we will follow common practice and reserve the term "Atlantic" for those regions of the Atlantic Ocean that are outside the AM.





et al., 2004). Through ventilation and overflow, the AM is one of the main regions linking the atmosphere and the deep World Ocean and the associated transport of atmospheric carbon dioxide into the deep ocean is critical for climate change on long time scales (Sabine et al., 2004).

The inflowing water from the Atlantic that does not return as overflow mixes with the Pacific inflow and leaves the AM through the Canadian Archipelago /Denmark Strait and the upper western Fram Strait as cold and relatively fresh "surface outflow" (Curry et al., 2014; de Steur et al., 2017).

Exchanges between the AM and the rest of the World Ocean can therefore be grouped into three types of flow that play important, but different, roles in the ocean and climate systems: *"inflow"* of water from the Atlantic and Pacific into the AM, *"overflow"* of dense water at depth from the AM into the Atlantic, and *"surface outflow"* in the upper layers into the Atlantic (Figure 2). In addition to these oceanic exchange flows, freshwater enters the AM as runoff, Greenland meltwater discharge and through net precipitation (Aagaard and Carmack, 1989; Serreze et al., 2006). Ice exports are not considered here as the volume transports are low (same citations).

The important role of the AM in the World Ocean circulation and global climate has been recognized for a long time. There have been many attempts to quantify the AM-exchanges and establish a budget for the AM since the pioneering attempt by Worthington (1970). Only recently, however, has the observational coverage become sufficiently comprehensive and reliable that a consistent budget may be determined with confidence.

The flows into and out of the AM are an integral part of the AMOC, which is projected to weaken during the 21$^{st}$ century (IPCC, 2013), and we discuss whether the observations show any indication of this.

In terms of area and volume, the AM is dominated by the Arctic Ocean, for which a budget was proposed by Beszczynska-Möller et al. (2011). Much of the water mass transformation and recirculation within the AM occurs, however, in the Nordic Seas (Mauritzen, 1996). A budget for the whole of the AM, which we try to establish here, can therefore be expected to be qualitatively different from a purely Arctic Ocean budget.

In the following sections, we first list the main features and observational systems for each individual exchange branch and the data sets that we use. The combined results of these data are given in Sect. 3 with the main focus on multi-year average transports, seasonal variations, and long-term trends. These results are discussed in Sect. 4, where we initially try to assess whether the combined data set is consistent – e.g. do the combined average transports and their seasonal variations conserve mass. After that, we discuss what is perhaps the most important outcome of this study: are the total flows into and out of the AM strengthening, weakening, or stable over the time period covered by our observations? The manuscript ends with Sect. 5 where we present our main conclusions and recommendations.

## 2 The exchange branches and their observing systems

In this section, we outline the main features of each individual exchange branch and of the observational systems used to quantify and monitor these exchanges. Following tradition (e.g. Hansen and Østerhus, 2000), we group them into the four





categories illustrated in Figure 2. The distinction between overflows and surface outflows is difficult, especially in the Denmark Strait where they flow together, and will be discussed later (Sect. 2.2.1 and 2.3.2). Here, we use the well-established criterion: $\sigma\theta > 27.8$ kg m$^{-3}$ to define overflow (e.g. Dickson and Brown, 1994).

The observational evidence from the individual exchange branches is highly variable. For some branches, we have time series of monthly averaged volume transport spanning more than two decades, although in some cases with gaps. For other branches, the time series are much shorter, or the observational evidence may be barely sufficient to provide one number for the average transport without any yielding any information on temporal variations.

To keep the text within readable limits, the descriptions provided in this section do not give complete information about each individual branch. Also, the variability in physical aspects as well as in published information makes it difficult to describe all the branches in a similar and consistent manner. Instead, our aim has been to provide enough information to place each branch as a part of the whole exchange system and describe the observing methodology. For each branch, we, list a few key references for access to more detailed information. Where essential details are not available in the literature, we have added information in the supplementary document.

## 2.1 Inflows

Most of the water entering the AM comes from the Atlantic Ocean (Atlantic inflow). The three main Atlantic inflow branches pass through the deep passages across the Greenland-Scotland Ridge (Figure 3), which for historical and logistical reasons are discussed separately. The remaining Atlantic inflows that we also have to consider are the inflows over the Scottish shelf and through the English Channel. Here we combine these two flows into a "European Shelf Atlantic inflow". Additionally, water from the Pacific Ocean (Pacific inflow) enters the AM through the Bering Strait.

### 2.1.1 Denmark Strait Atlantic Inflow (DS-inflow)

Denmark Strait, between Greenland and Iceland is about 300 km wide with a sill depth of 630 m. Within the strait, Atlantic water flows towards the Iceland Sea mostly over the Icelandic shelf but west of that, cold, low salinity Polar water exits the strait to the Irminger Sea as the surface outflow of the East Greenland Current. Over the deepest part of the strait, Denmark Strait overflow water also flows towards the Irminger Sea. The Atlantic inflow passes northwards with the surface Irminger Current along the west coast of Iceland. When it reaches Denmark Strait it splits into two branches with most of the water not flowing through the strait but flowing west across the Irminger Sea towards Greenland and subsequently southwestwards along the East Greenland continental slope. The other branch flows through Denmark Strait into the Iceland Sea and continues onto the North Icelandic shelf where it flows eastwards along the shelf as the North Icelandic Irminger Current (Stefánsson, 1962).

The method used for calculating the water mass composition on the Hornbanki section (H section, Figure 3) and the transport of Atlantic water is described in detail by Jónsson and Valdimarsson (2005) and Jónsson and Valdimarsson (2012). CTD (conductivity-temperature-depth) profiles from the Látrabjarg and Kögur standard sections that have typically



been sampled 4 times annually for the period when moorings were present on the Hornbanki section are used (L and K sections respectively, Figure 3). A station on the L section that always lies within the Atlantic water flowing northwards and a station on the K section that is within the Polar waters of the East Greenland Current are combined with temperature measurements from the Hornbanki mooring array (H section, Figure 3) to calculate the water mass composition at the H section, assuming

that it is a mixture of Atlantic and Polar waters. The current meter records from the H section are then used to calculate the transport of Atlantic water to the AM through Denmark Strait. The current meter measurements started in 1994 and have been maintained and made more extensive since then (Jónsson and Valdimarsson, 2012). In 1999 the array was extended from one mooring to three moorings and in 2012 a mooring was added north of the previous moorings. From 1994 to 2009, velocity at the H section was measured with single-point current meters, but starting in 2009, velocity measurements have been made

mostly with Acoustic Doppler Current Profilers (ADCP). There are several gaps in the individual current meter records probably due to fishing activity in the area and occasional icebergs, but the transport record is continuous since some of the moorings have always been recovered.

The time series for DS-inflow volume transport used in this study consists of monthly averages from October 1994 to December 2015.

### 2.1.2 Iceland-Faroe Atlantic Inflow (IF-inflow)

Between Iceland and the Faroes, the Iceland-Faroe Ridge has a sill depth around 480 m close to the Faroes, but is deeper than 300 m over much of its extent. Across this ridge, there is an inflow of Atlantic water to the Nordic Seas in the upper layers (IF-inflow), whereas (southward flowing) overflow water crosses the ridge in the opposite direction at depth. Both exchanges

occur over most of the length of the ridge, but likely with large temporal and spatial variations (Tait et al., 1967; Meincke, 1983; Perkins et al., 1998; Rossby et al., 2009). Due to the high spatial and temporal variability of these exchanges, a monitoring array located on the ridge that could generate time series of IF-inflow volume transport would need to be very extensive and would be vulnerable to the intensive fishing activity. This has not been attempted.

Instead, monitoring has been established on a section (the N-section, Figure 3) east of the ridge   where the inflow

crossing the ridge is focused into a relatively narrow boundary current, the Faroe Current.   This current flows eastwards north of the Faroes, bounded on the north side by the Iceland-Faroe Front (Tait et al., 1967;  Hansen and Meincke, 1979; Read and Pollard, 1992). The N-section has been sampled 3-4 times annually by  CTD cruises since the late 1980s. Since 1997, this has been complemented by an array of moored  ADCPs, deployed below the extent of fishing gear or in trawl-protected frames on the bottom. Based on  the combined ADCP and CTD data, Hansen et al. (2003) derived average estimates and time

series of  volume transport for the IF-inflow, representing the Atlantic water crossing the ridge.

The volume transport based solely on in situ observations was found to be well correlated with  the sea level tilt on the section derived from altimetry data (Hansen et al., 2010), and a new algorithm was  developed which combines data from altimetry and in situ observations (Hansen et al., 2015). Based on  this, the time series for IF-inflow volume transport used in this study consists of monthly averages from   January 1993 to December 2015.





### 2.1.3 Faroe-Shetland Atlantic Inflow (FS-inflow)

The gap in the Greenland-Scotland Ridge between the Faroes and Scotland is called the Faroe-Shetland Channel (FSC). The deepest part of the channel is deeper than 1000 m. Water of Atlantic origin usually fills the upper layers down to 400 –

500 m across the whole channel, but a significant fraction originally crossed the ridge north of the Faroes, entered the Faroe Current, and bifurcated into the FSC, where it flows southwestwards along the Faroe side of the channel, Figure 3 (Helland-Hansen and Nansen, 1909; Meincke, 1978; Hátún, 2004, Berx et al., 2013). Most of this water is believed to recirculate within the channel and join the direct inflow continuing into the Norwegian Sea (Hansen et al., 2017).

Regular hydrographic surveys along standard sections crossing the channel have been carried out for more than a

century (Tait, 1957; Turrell, 1995) and, since the 1970s, these have been complemented with current meter moorings and other instrumentation (Gould et al., 1985; Dooley and Meincke, 1981; Rossby and Flagg, 2012; Berx et al., 2013). In this study, we use data from the only long-term transport monitoring effort (Østerhus et al., 2001), consisting of CTD profiles and moored ADCP time series along a standard section (the Munken – Fair Isle section, labelled the M-section in Figure 3) starting in 1994. The recirculation of Atlantic water and intensive meso-scale activity (Sherwin et al., 1999; 2006; Chafik,

2012) complicate the calculation of volume transport. By combining the in situ observations with data from satellite altimetry, Berx et al. (2013) generated a time series of volume transport of the FS-inflow with monthly estimates from January 1993 to September 2011, here extended to December 2015.

The time series generated by Berx et al. (2013) represents the Atlantic water flow between the shelf edges on both sides of the channel. On the Faroe shelf, northwest of the shelf edge boundary of the channel, there is a flow between the islands

and the shelf edge, which generally is directed southwestwards. Most of this is considered to belong to a quasi-closed shelf circulation around the Faroes (Larsen et al., 2008) and therefore is not advected into the AM. This shelf circulation is not included in the IF-inflow as it passes eastwards north of the Faroes (Hansen et al., 2003) and should therefore not be included in the FSC either. For the continental shelf region southeast of the FSC monitoring section, there is, on the other hand, an Atlantic inflow, which is not recirculated around the UK. That contribution is discussed in the next section, section 2.1.4.

### 2.1.4 European Shelf Atlantic inflow (ES-inflow)

The European Shelf (ES)-inflow is the inflow of Atlantic water between the southeastern boundary of the Faroe-Shetland Channel monitoring system and the European continent. The previously discussed (Section 2.1.3) Atlantic water flow through the channel – the FS-inflow – has been monitored on a section (the M-section in Figure 3) that terminates at a point just

inside the shelf edge on the Scottish shelf with bottom depth ~150 m (Berx et al., 2013), (bottom right extent of white line on Figure 4). Between this point and Orkneys, there is a distance of ~125 km (which we call here the Scottish shelf), through which there may be appreciable flow. Unfortunately, this has not been systematically monitored and observationally based estimates of its volume transport seem difficult to find.



Despite this lack of observational evidence, it seems clear that the average volume transport over the Scottish shelf must at least be equal to the average volume transport of the Fair Isle Current that passes into the North Sea through the gap between Orkneys and Shetland – the Fair Isle Gap (Figure 4). This current was estimated by Turrell et al. (1990) to have an average transport of 0.13 Sv. Their observations only covered a few months, however, and Hill et al. (2008) have updated this value to 0.4 Sv, based on a combined observational and modelling effort.

This value may thus represent a minimum average volume transport over the Scottish shelf, but some of the water over the shelf may continue northeastwards to flow west of Shetland rather than passing through the Fair Isle Gap. Again, there is little observational evidence, but some information may be gained from measurements by a ferry-mounted ADCP (Rossby and Flagg, 2012). The focus of the ADCP data acquisition was on larger scales, but from their graphs and updated graphs reported by Childers et al. (2014), we estimate an additional ~0.1 Sv of water flowing into the AM, giving a total average volume transport of 0.5 Sv over the Scottish shelf inside of the M-section.

The flows over the Scottish shelf and through the English Channel include less saline water from coastal areas upstream in addition to the more oceanic component. Thus, the term "Atlantic" may be somewhat misleading but, for our purpose, it is the total volume transport rather than the characteristics of the water that is important. These coastal water masses are therefore included in the ES-inflow.

From in situ observations, there is little evidence about the variations of volume transport, but satellite altimetry may be used for that purpose as long as we can assume geostrophy, which works well for the neighbouring FS-inflow (Berx et al., 2013). As elaborated on in the supplementary document, we have therefore combined the established average transport value with Sea Level Anomalies (SLA-values) from altimetry to generate monthly time series of ES-inflow with the additional assumption of barotropic flow. This assumption probably leads to transport variations that are too high, but they are still low in absolute terms and should not have much influence on the overall picture.

In addition to the flow over the Scottish shelf, there is also an inflow of Atlantic water through the English Channel, which according to the observations reported by Prandle (1993) has an average volume transport of ~0.1 Sv. Altogether, we will therefore use a value of $(0.6 \pm 0.2)$ Sv for the average volume transport of the ES-inflow where the uncertainty value is estimated from the limited observational evidence.

### 2.1.5 Bering Strait Pacific inflow (BS-inflow)

The Bering Strait is a narrow (width ~85 km) and shallow (sill depth ~50 m) strait connecting the Pacific and Arctic oceans (Figure 5). Since 1990, year-round measurements have been maintained in the strait almost without interruption, typically at 2-3 sites (Figure 5) located within one or both of the two channels of the strait (sites A1 and A2), and typically also at a mid strait site, A3, slightly to the north, at a location found to give a useful average of the flows from the two channels (see. e.g. Woodgate et al., 2015 for discussion). In 2001, a mooring (A4) was added in the eastern side of the eastern channel to monitor the warm, low-salinity Alaskan Coastal Current (ACC) present seasonally (Woodgate et al., 2015).

In the 1990s, velocity at the mooring sites was measured mostly by single-point current meters, but since 2007,



velocity measurements have been made predominantly with ADCPs. Based on the observed dominantly barotropic and spatially homogeneous nature of the flow (away from the ACC), volume transport is calculated by multiplication of velocity and cross-sectional area for the strait (Woodgate et al., 2018).

Over the period of monitoring (1990 to present), there has been a statistically significant increase in annually averaged volume transport from 0.8 Sv in the beginning of the period (Roach et al., 1995) to ~ 1.2 Sv by the end (Woodgate et al., 2018). Here, we use the monthly mean volume transports from August 1997 to December 2013.

## 2.2 Overflows

The only deep connections between the AM and the rest of the World Ocean are the gaps in the Greenland-Scotland Ridge
and only through these gaps do we find the flows of dense water from the AM that are generally characterized as "overflow". In the literature, various criteria have been used to define overflow – either in terms of temperature or density. In this study, we use the most common definition: $\sigma_\theta > 27.8$ kg m$^{-3}$ (e.g. Dickson and Brown, 1994). We also follow common practice (e.g. Hansen and Østerhus, 2000) to group the overflow into four different branches (Figure 6).

### 15  2.2.1 Denmark Strait Overflow (DS-overflow)

About half of the dense overflow waters from the Nordic Seas enter the North Atlantic through Denmark Strait, where the DS-overflow becomes one of the major sources of NADW (e.g. Dickson and Brown, 1994). The overflow plume crossing the passage between Iceland and Greenland is generally found at a depth below 250 m, although close to the Icelandic shelf warm and saline Atlantic water frequently occupies the passage down to the bottom (Mastropole et al., 2017).

The width of Denmark Strait which is deeper than 350 m covers a distance of 60 km only. Here, the overflow plume is most intense with downstream velocities exceeding 1 m s$^{-1}$ and near-bottom temperatures below zero. Mesoscale eddy activity is well documented, and occurs with periods of 2-10 days (Ross, 1984; Käse et al., 2003; Fischer et al., 2015) whereas seasonal variability is small and no significant long term trends have been found so far (Jochumsen et al., 2012). Moored instrumentation for current profile measurements (ADCPs) have been installed in this part of the passage (the
L-section in Figure 6). The standard deployment consists of two moorings, one at 650 m depth at the deepest part of the sill of the strait, the other 10 km further towards Greenland at 570 m depth. These positions cover the overflow current core, but a large volume of dense water on the Greenland shelf region is not accounted for.

      Velocities on the shelf are small, but the distance to the coast of Greenland is still more than 250 km, where some dense water is transported southward (de Steur et al., 2017). In earlier publications, this transport was inferred from a model
and added to the transport calculations obtained by the moorings (Macrander et al., 2005; Jochumsen et al., 2012). In 2014/2015, however, an experiment was made with five moorings on the L-section, from which a new algorithm was developed to derive volume transport from the historical ADCPs observations (Jochumsen et al., 2017). The monthly averaged DS-overflow transport values used here are based on this algorithm and extend from May 1996 to December 2015,

although with gaps.

A quality check on this new time series is provided by the experiment reported by Harden et al. (2016) with a dense mooring array on the K-section (Figure 6) lasting from September 2011 to July 2012. For the overlapping period (336 days), our data set based on Jochumsen et al. (2017) has an average transport of 3.1 Sv, whereas Harden et al. (2016) find 3.5 Sv. Considering the uncertainties reported (±0.5) and possible water mass transformations between the two sections, this comparison is encouraging (Jochumsen et al., 2017).

### 2.2.2 Iceland Faroe Ridge Overflow (IF-overflow)

Overflow across the Iceland-Faroe Ridge was identified more than a century ago (Knudsen, 1898) and it has long been known that it may occur at many locations along the ridge (Hermann, 1967; Meincke, 1983). From the results of the Overflow '60 expedition, Hermann (1967) estimated a total volume transport of 1.1 Sv for the IF-overflow. Based on moorings and hydrographic observations, Perkins et al. (1998) estimated at least 0.7 Sv overflow close to Iceland, and Beaird et al. (2013) used measurements from autonomous Seagliders to find a minimum of 0.8 Sv for the total overflow across the ridge. Observationally-based information of temporal variations or time series of total IF-overflow have not been published, however.

The Iceland-Faroe Ridge may conveniently be divided into two parts at the latitude of 63°N, figure 6. Across the southern (Faroese) part, the overflow is considered to be intermittent (Østerhus et al., 2008) and from their extensive Seaglider experiment, Beaird et al. (2013) estimated an average volume transport of that part of the overflow of 0.3 Sv with an uncertainty (estimated from their figures) almost as high.

Across the northern (Icelandic) part, the overflow has generally been thought to be more persistent (Østerhus et al., 2008), especially the overflow through the northernmost passage across the ridge, the Western Valley (Figure 6). This is partly from theoretical arguments and partly from observations of a strong and persistent bottom current downstream from the Western Valley that seems to have been generated by IF-overflow (Perkins et al., 1998; Olsen et al., 2016). Measurements within the Western Valley have not, however, shown any clear evidence of strong overflow (Perkins et al., 1998; Beaird et al., 2013) and based on a dedicated field experiment from August 2016 to May 2017, Hansen et al. (2018) argue that the long-term average overflow transport through the Western Valley is less than 0.1 Sv.

Following these recent results, we use the value 0.4 Sv for the average transport of IF-overflow with an uncertainty of ±0.3 Sv. This value for the average transport may seem small when the bottom current downstream of the Western Valley is taken into account (Perkins et al., 1998; Olsen et al., 2016), but the volume transport of this current is not well constrained by observations and neither are its origin and on-route entrainment of Atlantic water. From bottom temperature measurements (Olsen et al., 2016), it also seems unlikely that much of this water would fulfil the criterion for overflow. Seasonal and long-term variations of the IF-overflow cannot be addressed with the observational material available.



### 2.2.3 Faroe Bank Channel overflow (FB-overflow)

The Faroe Bank Channel is the deepest passage across the Greenland-Scotland Ridge with a sill depth of 840 m. The bottom layer in this channel is continually dominated by cold overflow water that flows over the sill with core speed usually exceeding 1 m s$^{-1}$ out into the Atlantic (Hermann, 1959; Borenäs and Lundberg, 1988; Saunders, 2001; Hansen and Østerhus, 2007; Hansen et al., 2016).

Since the early estimates by Hermann (1967) and Sætre (1967), several transport estimates for the FB-overflow have been published. Here, we use the most comprehensive data set consisting of data from long-term ADCP moorings on the V-section (Figure 6), combined with other moored instrumentation and regular CTD cruises (Hansen et al., 2016). The primary time series generated from these observations, the "kinematic overflow", has an average volume transport of

2.1 Sv, which, however, includes 0.2 Sv of water less dense than the established criterion ($\sigma\theta \geq 27.8$ kg m$^{-3}$). For our purpose, the time series has therefore been converted by multiplying the values with a fixed ratio of (2.1-0.2)/2.1. The series contains monthly averaged volume transport from December 1995 to December 2015, although with gaps during the annual servicing periods.

### 2.2.4 Wyville Thomson Ridge Overflow (WT-overflow)

The Wyville Thomson Ridge has a sill depth of around 600 m with intermittent overflow of dense water both at the deepest point at the centre of the ridge and at the far west of the ridge (Ellett and Roberts, 1973; Sherwin et al., 2008). This flow, the WT-overflow, is channelled by topography into the Ellett Gully before entering the Rockall Trough to the south. The flow through the Ellett Gully has primarily been monitored by ADCP moorings but also by a CTD section (the W-section in

Figure 6).

The time-varying nature of WT-overflow necessitates the combination of volume transports with proportions of Faroe Shetland Channel Bottom Water (FSCBW) in order to produce a transport comparable to other overflow timeseries (Sherwin et al., 2008). In this method, the volume transport through the Ellett Gulley, as measured by the moored ADCP, is weighted by the proportion of FSCBW in the water column, calculated from linear mixing between FSCBW (defined as

having a temperature of 0°C) and Atlantic Water (defined as having a temperature of 8.5°C). The method assumes temperature decreases linearly from the depth of the 8.5 °C isotherm to the seabed, and that the isotherms are horizontal. Sensitivity tests suggest that the error associated with these assumptions are less than ± 20 % (± 0.04 Sv). The timeseries of WT-overflow used in this study is based on this method and consists of monthly averages from May 2006 to May 2013, although there is a data gap from June 2009 to May 2011 due to instrument loss.

The definition of FSCBW is denser than our criterion for overflow water (27.8 kg m$^{-3}$) and thus, 0.2 Sv is a lower bound for the volume transport. Recently, Johnson et al. (2017) reported a mean of 0.3 Sv for the WT-overflow denser than 27.65 kg m$^{-3}$ further south in the central Rockall Trough. Although some WT-overflow may have exited the Rockall Trough via the northern banks, we consider an average value for the WT-overflow transport between 0.2-0.3 Sv to be



appropriate. We therefore use the time series of FSCBW transport based on the method of Sherwin et al. (2008) and attach an uncertainty of ±0.1 Sv.

## 2.3 Surface outflows

In addition to the overflow of dense water through the deep passages across the Greenland-Scotland Ridge, the AM also exports water that is less dense and remains in the upper layers. The flow of these water masses is denoted as "surface outflow" or just "outflow" and it may be seen as two separate branches passing on either side of Greenland.

### 2.3.1 Canadian Archipelago surface outflow (CA-outflow)

The Canadian Archipelago (CA) is a collection of numerous islands separated by narrow sounds. Through these sounds and through the Nares Strait, separating the CA from Greenland, there is a net flow of water from the Arctic Ocean towards the Labrador Sea (Figure 7). Measurements of these flows are difficult due to ice, strong tidal currents, recirculation, and proximity to the magnetic pole. Nevertheless, volume transport has been estimated from observations at several locations (Melling et al., 2008).

15           Davis Strait connects Baffin Bay to the Labrador Sea and has a sill (640 m depth) that limits deep exchanges between the two. Outflow from Baffin Bay through Davis Strait carries inputs from the integrated CA outflows as well as northward inflow along the West Greenland shelf and slope that has been modified during transit in Baffin Bay. Transport through Davis Strait has been monitored using a mooring array south of the sill that includes velocity, temperature, and salinity measurements from 15 moorings spanning the full width (330 km) of the strait accompanied by autonomous Seaglider surveys

(Curry et al., 2014).

          Transport through Davis Strait is used to represent the CA-outflow in this study. We use monthly averaged volume transports from October 2004 to September 2010. There is a small component of the Arctic Ocean outflow that bypasses Baffin Bay and flows through Fury and Hecla Strait (Figure 7). Its volume transport is not well constrained but has been estimated to be less than 0.1 Sv (Straneo and Saucier, 2008). It will not be included here.

### 2.3.2 Denmark Strait surface outflow (DS-outflow)

The surface outflow through the Denmark Strait is difficult to monitor. At times, it may flow through a large part of the width of the strait, requiring a wide and dense mooring array while the component flowing over the East Greenland shelf is inundated with icebergs that are very destructive to moored instrumentation. It therefore comes as no surprise that

observation-based transport values of the DS-outflow have been late to arrive.

          The values used here are mainly based on the experiment described in Sect. 2.2.1 with a dense mooring array along the K-section (Figure 6) from September 2011 to July 2012 (Harden et al., 2016). There, the focus was on the dense-water component ($\sigma\theta > 27.8\,\mathrm{kg\,m^{-3}}$), but the transport of the less dense water masses ($\sigma\theta < 27.8\,\mathrm{kg\,m^{-3}}$) could also be derived from



the observations as reported by de Steur et al. (2017). They estimated the average transport of this upper-ocean component to be 1.8 Sv towards the southwest with an uncertainty on the order of ±0.5 Sv. This value does not, however, cover the East Greenland shelf region adequately.

To amend this, we add data from additional inshore moorings on the K-section from 2012 to 2014 reported by de
Steur et al. (2017). From these additional data, monthly averages of the transport over the shelf can be generated, and we add these to the monthly averages from the 2011-2012 experiment (Figure 9b in de Steur et al., 2017). In this way, we obtain a time series of 11 months from September 2011 to July 2012, which should include the total surface outflow through the Denmark Strait. The validity of this approach is of course dependent on the stationarity of the seasonal cycle over the shelf, which is questionable, but the modification due to the addition of the 2012-2014 inshore moorings is small.

On this basis, we have estimated a value of 2.0 Sv for the average DS-outflow. This value is composed of two non-concurrent contributions, the dominant of which was based on only 11 months of observations. It must therefore be treated with caution, as must the seasonal variation indicated by the data, which shows a pronounced winter-intensification of the DS-outflow. A strong seasonality of the flow over the Greenland slope is, however, supported by more prolonged current measurements in this region (Jónsson, 1999). The transport of the East Greenland Current at 74°N was also found to
be subject to a large seasonal cycle related to the wind-driven gyre in the Greenland Sea (Woodgate et al., 1999).

## 2.4 Runoff and precipitation (Freshwater input)

In addition to oceanic inflows, water enters the AM by net precipitation (precipitation minus evaporation) and runoff from rivers as well as land-ice melting into the sea, which we consider collectively as "Freshwater input". Since the various
freshwater contributions have relatively small magnitudes, they are commonly reported in mSv (1 mSv = $10^{-3}$ Sv).

The freshwater budget of the Arctic Ocean was pioneered by Aagaard and Carmack (1989) and updated by Serreze et al. (2006) who reported a net precipitation of 65 mSv and a runoff of 102 mSv to the Arctic Ocean. Including also the Nordic Seas, Dickson et al. (2007) added 20 mSv of net precipitation and 34 mSv of runoff from the Baltic, the Norwegian coast, and Greenland. Another 9 mSv enter the Canadian Archipelago from Greenland according to Dickson et al. (2007).
This yields a total freshwater input to the traditional AM of 230 mSv, which we round to 0.2 Sv with an estimated uncertainty less than 0.1 Sv. Since we include also the North Sea in our definition of the AM (Figure 1), there are additional inputs, especially river runoff from Belgium, Netherlands, and Germany into the North Sea, but they are only a few mSv and too small to affect this value (Radach and Pätsch, 2007).

Most of the freshwater contributions exhibit strong seasonality. According to Serreze et al. (2006), the net
precipitation to the Arctic Ocean is more than twice as high in July as in March and river runoff to the Arctic Ocean has an even more pronounced seasonal variation. A similar, although less extreme, seasonal variation has been reported for river runoff to the Baltic (Bergström and Carlsson, 1994). Within the uncertainties generally applying to this study, it therefore seems safe to assume a seasonal variation of Freshwater input to the AM with an amplitude around 0.1 Sv and maximum



around July.

In addition to seasonal, there are also long-term variations and Haine et al. (2015) suggest that net precipitation and runoff to the Arctic Ocean and Canadian Archipelago were greater in the 2000s than for 1980-2000. The observational evidence for this is, however, weak and in any case within the quoted uncertainty. Thus, it will be ignored here.

## 3 Results

As described in the previous section, monthly transport values are available for almost all of the oceanic exchange branches into or out of the AM, although of highly variable duration and completeness (Table 1). The one exception is the IF-overflow that has not been systematically monitored and for which we only have estimated a typical or "average" transport value and

its uncertainty. For the Freshwater input, likewise, we only have an average value and a seasonal amplitude. In the following, we present the average transports, as determined from the various data sets (over differing time-periods), as well as their variations on seasonal and long-term time scales. Transport values are defined as positive into the AM and negative out of the AM.

### 3.1 Average volume transports

Combining all the inflow transports with the freshwater input, we get the total "***AM-import***", which has an average value of 9.1 Sv. Likewise, we can combine all the overflow transports with the surface outflow transports to an "***AM-export***" with an average value of -9.5 Sv. Hence, the export exceeds the import so that the average "***Net import***" (AM-import + AM-export) is -0.4 Sv. Combining the various uncertainty terms, this number has an uncertainty exceeding 1 Sv.

Thus, the imbalance in the average Net import is within the combined uncertainties even though the various

numbers in Table 1 are averaged over widely different periods. The most complete coverage is during a 6-year period October 2004 to September 2010, in which there are 53 months with data from all of the inflow branches, from the DS-overflow, the FB-overflow, and from the CA-outflow. The sum of the transport values for all of these branches in these months did not, however, differ substantially from the sum based on the full periods (Table 1).

### 25  3.2 Seasonal variation

Table 1 also lists the standard deviation of the monthly transport values for each individual branch and some branches are clearly more variable than others, especially when considering the ratio between standard deviation and average transport. Thus, the monthly standard deviation of the FS-inflow is almost twice that of the IF-inflow even though the IF-inflow has a higher average transport.

Some of this variability seems to derive from systematic seasonal variations as indicated in Figure 8, where we have compared seasonal variations during the most complete 6-year period. The inflow branches seem to have different seasonal variations (Figure 8a), with the IF-inflow, the FS-inflow, and the ES-inflow being strongest around the turn of the





year, whereas the BS-inflow and the DS-inflow are strongest in summer. For the overflow and surface outflow branches, the picture seems less clear (Figure 8b) and most of the export branches do not exhibit any clear seasonality.

To get a more complete impression of the seasonal variation, the monthly transport values for the five inflow branches in Figure 8a have been summed to give the total AM-import when the Freshwater input is neglected.
Subtracting the overall average, we get the seasonal import anomaly, which is shown as the red curve in Figure 9. Similarly, the blue curve in that figure shows the seasonal anomaly of the AM-export although note that this neglects the IF-overflow and missing months for the FB-overflow and DS-outflow had to be interpolated to get a complete seasonal coverage.

Combining the red and blue curves in Figure 9, we get the seasonal anomaly of the Net import for those branches that have been sufficiently well observed (black curve). It seems to indicate a maximum in November and minimum in
August with an amplitude on the order of 1 Sv. A more detailed discussion of this imbalance will be presented in Sect. 4.2, but it is worth emphasizing that this curve is not based on a very homogeneous data set. The inflow branches and the CA-outflow had no gaps in the selected period, but that was not the case for the overflow branches and our data for the DS-outflow only cover 11 months and they are outside the selected period.

## 3.3 Long-term variations

Only the five inflow branches and the two main overflow branches have been observed over sufficiently long periods to allow a meaningful investigation of possible long-term variations or trends. The FB-overflow has months missing for almost every year but for the other branches, annual averages may be computed for most of the years within the observing period. Based on these annual averages, Table 2 lists linear trends as calculated by linear regression of annual volume transport on
time.

Except for the BS-inflow, the trends in Table 2 are less than their confidence intervals, which are calculated without taking serial correlation into account (and thus these confidence intervals are likely to be underestimates of the real uncertainty in the trend). The exchanges between the AM and the Atlantic are therefore characterized by stability rather than change.

A more illustrative picture of the long-term variation is presented in Figure 10, which shows low-passed series generated by averaging all observed months (up to 36) in 3-year periods. For some branches, months were missing for some of the 3-year periods, but never more than 6 months. Thus, all the points in Figure 10 are averaged over at least 30 months. The curves in Figure 10 are consistent with Table 2 with only weak trends for most of the branches and relatively small variations.

The longest time series considered here are for the four Atlantic inflow branches and the two main overflow branches. From 1996 to 2015, the Atlantic inflow branches had almost complete coverage and the total volume transport of these branches had at most two months missing in every 3-year period. Thus the thick red line in Figure 11 should give a good representation of the variations of the total Atlantic inflow during these 18 years. The sum of the two main overflow





branches has less complete coverage, but the de-seasoned 3-year running mean (thick blue line in Figure 11) still should give an indication of the variation of this series.

Figure 11 shows the change in inflow/overflow relative to their late 1990s values. For both the total Atlantic inflow and the sum of the two main overflow branches, Figure 11 seems to indicate strengthening from the late 1990s to 2002 with little overall change after that. When taking the uncertainties (coloured areas on Figure 11) into account, the statistical significance of the apparent changes seems low, however, and the overall message is one of stability.

## 4 Discussion

The results presented in the previous section are from a wide and inhomogeneous set of observational systems. The first question to ask is therefore whether they are mutually consistent. From Table 1, the estimated AM-export is 0.4 Sv higher than the AM-import, but this imbalance is well within the uncertainties quoted in the table and in principle needs no further explanation. Whether to expect a zero imbalance in our data set is, however, not as obvious as might be thought and deserves discussion (Section 4.1).

Similarly, the Net import in our data set appears to have a non-zero seasonal variation (Figure 9) and we need to ask whether it is within acceptable bounds. The following sections 4.1 and 4.2 therefore address what constraints nature puts on the average value and seasonal variation of the Net import. Another problem is that the individual observational systems do not combine into a contiguous whole. This is discussed in Sect. 4.3. The implications of the apparent imbalances in our results for data quality are summarized in Sect. 4.4. In a very simplified picture, the AM may be seen as a double estuary with an estuarine as well as a thermohaline loop. In Sect. 4.5, we estimate the relative strengths of these two loops and their sources. After that, in Sect. 4.6, we address the important question: have the total flows into and out of the AM been weakening, strengthening, or remained stable within our observational period?

### 4.1 Constraints on the average AM-exchange budget

The ultimate criterion for a consistent exchange budget is mass conservation. When there is an imbalance between import and export, the total mass within the AM must change accordingly. If there were no density changes, the mass balance would be equivalent to volume balance (continuity). An imbalance of 0.1 Sv that was sustained for a year would then imply a sea level change around 20 cm on average over the whole AM. This is considerably more than available observations indicate for inter-annual sea level variations (Volkov and Pujol, 2012; Andersen and Piccioni, 2016), although observational evidence is missing for much of the Arctic Ocean. Basin-wide GRACE Ocean Bottom Pressure data suggest interannual trends between 2002-2006 of only a few cm (<5cm/yr) over the Arctic basin, and of varying sign (Morison et al., 2007), more evidence that a 0.1 Sv imbalance is unrealistic.

In reality, air-sea exchanges and mixing with runoff and other water masses induce density changes in the water between entering and leaving the AM, but they are too small to affect this calculation significantly. In addition





to this, the induced expansion and contraction of the water result in steric sea level changes within the AM that add to the mass-induced changes but, again, the inter-annual variations caused by this are considerably smaller than 20 cm in the areas reported (Mork and Skagseth, 2005; Andersen and Piccioni, 2016).

When averaging over a period of a year or longer, the imbalance between AM-import and AM-export therefore has
to be considerably smaller than 0.1 Sv. The imbalance we find in our observational estimates of the import/export are thus almost certainly due to the present limitations of the observational system.

## 4.2 Constraints on the seasonal AM-exchange variations

For the seasonal variation in transports, mass conservation must again be required but now the implications are more
intricate. As a framework for the discussion, consider a model, in which the Net import anomaly (Net import minus its temporal mean), $Q(t)$, varies sinusoidally with time, $t$:

$$Q(t) = Q_0 \cdot \cos\left[\frac{2\pi}{T} \cdot (t - \tau_Q)\right] \qquad (1)$$

where $Q_0$ and $\tau_Q$ are the seasonal amplitude and time of maximum for the Net import, respectively, and $T$ is one year. Initially, we furthermore assume incompressibility so that there are no density changes and no steric sea level variations. In that case, continuity requires that the sea level height anomaly (sea level height minus its temporal mean) averaged over the whole of the AM, $H(t)$, fulfils

$$A\frac{dH}{dt} = Q(t) \qquad \Rightarrow \qquad H(t) = H_0 \cdot \cos\left[\frac{2\pi}{T} \cdot (t - \tau_H)\right] \qquad (2)$$

where $A$ is the surface area of the AM and the seasonal amplitude of $H(t)$, $H_0$, as well as the time of sea level maximum, $\tau_H$, are given by:

$$H_0 = \frac{T}{2\pi A} \cdot Q_0 \quad \text{and} \quad \tau_H = \tau_Q + \frac{T}{4} \qquad (3)$$

In reality, the assumption of incompressibility is not valid but this problem can be circumvented by subtracting steric seasonal variations from $H(t)$ before calculating $H_0$ and $\tau_H$. If we can determine $H_0$ and $\tau_H$ from observations, we can therefore
estimate what values $Q_0$ and $\tau_Q$ should have.

In the Nordic Seas, there is fairly good observational evidence of seasonal sea level variations from satellite altimetry. In this region, Mork and Skagseth (2005) found that a sinusoidal variation typically explained 40 – 50 % of




the total variance. Over the deep parts, maximum sea level occurred around September with amplitudes between 4 and 8 cm. They furthermore found that the steric component was in phase with the observed total variation and typically contributed around 40 %. These results were validated by Volkov and Pujol (2012) who compared the altimetry data with tide gauge records and also extended the region to include the Barents and Kara seas. Except for near-coastal areas, it seems

that when corrected for steric variation, the average value for $H_0$ in this region does not exceed 5 cm and maximum sea level is in autumn.

In the open Arctic Ocean, ice cover and lack of satellite coverage put severe limits on our knowledge of sea level variations but recently, Andersen and Piccioni (2016) have reported an analysis of sea level variation in the region from 66°N to 82°N, which supports the value of 5 cm as a maximum for $H_0$ in the AM as a whole. Similarly, Peralta-Ferriz and

Morison (2010), find, from GRACE data, a seasonal cycle within the Arctic of range ~5 cm. Combining this information with Eq. (3), we conclude that in nature, the Net import anomaly, $Q(t)$, is maximum in summer and its amplitude, $Q_0$, does not exceed 0.2 Sv.

The seasonal anomaly of the Net import derived from our data set (black curve in Figure 9) is not very consistent with this. In that figure, the seasonal anomaly of Freshwater input is missing but that should have little effect on the imbalance.

Also missing from Figure 9 is the seasonal anomaly of the IF-overflow but, again, this is not likely to explain the inconsistency between our seasonal Net import anomaly and the seasonal sea level variations in the AM from the literature. Thus, our much greater seasonal anomaly of >1Sv again reflects the limitations of the observational system, as we discuss next.

## 4.3 The contiguity of the combined observational system

By including the ES-inflow, we have tried to fill the largest hole in the observational system, but the system is still not completely closed. Through the two shallow passages, the Bering Strait and the Canadian Archipelago, the flow system is comparatively straightforward (BS-inflow and CA-outflow). Through the deep passages across the Greenland-Scotland Ridge, in contrast, there are flows both into and out of the AM and this creates problems for the contiguity of the combined

system.

One of these problems is that the import branches and the export branches have not generally been monitored on the same sections. This implies that some water may be counted both in the import series and the export series or may be missed altogether. This is especially a problem in areas with high mesoscale activity like the Iceland-Faroe region (Hansen and Meincke, 1979; Willebrand and Meincke, 1980; Allen et al., 1994).

Another problem is that a monitoring section may have other water passing through the section in addition to the water that is to be monitored. This is the case for all the passages across the Greenland-Scotland Ridge. Usually, hydrographic characteristics are used to distinguish the wanted water mass from the others (Jónsson and Valdimarsson, 2012; Berx et al., 2013; Hansen et al., 2015) but this may introduce considerable uncertainty, especially over periods with changing hydrographic





properties. This problem is exacerbated when different criteria are used for import and export branches through the same passages. Thus, the criteria for identifying Atlantic water crossing the Greenland-Scotland Ridge have generally been different from the criterion used to define overflow water flowing through the same passages.

The ambiguities associated with using hydrographic characteristics to identify the water to be monitored are the origin of a large fraction of the uncertainties listed in Table 1. This is, however, more like an unknown bias than a randomly varying observational error and transport variations ought not to be so much affected by this bias as long as the hydrographic properties of water masses do not change too dramatically.

## 4.4 The exchange budget of the AM and data quality

From Sect. 4.1 and Sect. 4.2, it is clear that neither the average transports nor the seasonal variations that we have observed combine into an exchange budget that is balanced within the constraints put by nature. For the average transports, the observed imbalance is well within the combined uncertainties of the various branches, but much of it may also be explained by the lack of contiguity in the observational system between Greenland and Scotland.

Thus, a substantial part of the uncertainties quoted for the average transports of the DS-inflow, the IF-inflow, and the FS-inflow is associated with the ambiguities involved in distinguishing the Atlantic water component from other water masses that do not derive directly from the Atlantic. Also, these branches have not been monitored on the same sections as the branches of overflow and surface outflow through the same passages. We should therefore not expect a perfect balance between average AM-import and AM-export. How much of the observed imbalance can be explained by this is difficult to estimate, but it may well be a substantial part.

As noted above, much of the uncertainty introduced in this way ought to be in the form of an unknown bias rather than a randomly varying observational error. We therefore expect the temporal transport variations to be less uncertain than the average values. With that in mind, the observed imbalance in the seasonal variation (Figure 9) is perhaps more problematic than the imbalance in average values. The data set, on which Figure 9 is based, is not very homogeneous, however. The five inflow branches had full coverage for the selected 6-year period (October 2004 to September 2010), as did the CA-outflow, but all the other branches had missing months in the data set.

The worst coverage is for the DS-outflow, for which there were no transport values in the 6-year period. For this branch, we only have 11 months of data and even those months did not cover the full DS-outflow (Sect. 2.3.2). The data may also be affected by the passage of a large anticyclone through the Denmark Strait in November 2011 (de Steur et al., 2017), perhaps helping to explain the large imbalance in Figure 9 for November.

During the selected 6-year period, the WT-overflow also had data gaps totalling 35 months and the DS-overflow had a gap of 10 months. For the FB-overflow, there was no year with complete coverage during the month of June (Figure 8b). For the June value in Figure 9, the FB-overflow was therefore interpolated, which may help explain the large imbalance for that month.

It therefore seems likely that the apparent seasonal imbalance in Figure 9 to a large extent may be explained by the





lack of data coverage for most of the export branches during the selected 6-year period. If that is correct, then our time series for the AM-import may be more accurate than indicated by the combined uncertainties in Table 1. Certainly, the seasonal variation of the AM-import in Figure 9 appears highly consistent and a sinusoidal seasonal variation explains 85% of the variance in the monthly averaged AM-import anomaly.

Combining the uncertainties for the AM-import branches in Table 1 quadratically, as commonly done, should therefore give a conservative estimate of the overall uncertainty and we conclude that the average AM-import for our observational period was (9.1±0.7) Sv. The AM-import furthermore seems to have a consistent seasonal variation with an amplitude of 0.9 Sv and maximum import in October. It must be emphasized, however, that these values depend on the definitions for the individual inflow branches, especially the Atlantic inflows.

For the AM-export, the data coverage is worse and uncertainties remain high. It might be argued that the average AM-export should equal the average AM-import in magnitude, given the constraints put by nature (Sect. 4.1), but that would require a contiguous observational system, which is not the case (Sect. 4.3). Nevertheless, our results do allow a consistent budget within reasonable uncertainties as illustrated in Figure 12, where we have updated and completed the Atlantic water budget presented by Hansen et al. (2008) (their Figure 1.15) to a budget for the whole of the AM.

The non-contiguity of the combined observational system may perhaps also affect the seasonal variation of the Net import (Figure 9), but probably less than it affects the average balance. From this and the discussion in Sect. 4.2, we would therefore expect the AM-export to have a seasonal amplitude close to 1 Sv and be strongest (most negative) around October. From our knowledge about the other export branches (Figure 8b), most of this seasonality would have to come from the DS-outflow, i.e., the estuarine loop, which is consistent with the available knowledge (Jónsson, 1999; de Steur et al., 2017), but will have to await future observational efforts for confirmation.

For the purpose of this study, it might have been advantageous if the monitoring of the various import and export branches in the Greenland-Scotland region had been better coordinated with identical monitoring sections for import and export branches. For other purposes, this might not be the case, however. Thus, the effects of the Atlantic inflow branches on conditions in the AM may be better monitored somewhat downstream from the intensive mixing areas over the Greenland-Scotland Ridge.

## 4.5 The AM as a double estuary

It is well known (e.g. Rudels, 2010) that the AM may be seen as a double estuary with both an estuarine and a thermohaline circulation. In Figure 12, the Atlantic inflow is split into two parts by two circulatory loops that feed the overflow and the surface outflow, respectively. The water mass transformations associated with the formation of overflow water occur in the Nordic Seas and the shelf seas north of Eurasia (Hansen and Østerhus, 2000). The Atlantic inflow is cooled by the atmosphere and freshened by mixing with freshwater. The low-density Pacific water enters through the Bering Strait and most of it leaves through the Canadian Archipelago (Rudels et al., 2004) although Bering Strait waters are also found in Fram Strait some years (Falck, 2001). Most of this low-density water mass does therefore not pass through





the overflow-formation areas and is not likely to contribute appreciably to overflow production (although it may affect water transformations outside the AM, e.g., in the Labrador Sea). To a first approximation, overflow water may therefore be considered a mixture of Atlantic water and freshwater in a mixing ratio of 99:1, based on the typical salinities of Atlantic water (~35.3; González-Pola et al., 2018) and overflow water (~34.9; Jochumsen et al., 2012; Hansen and

Østerhus, 2007).

This budget implies that around 70% of the total Atlantic inflow to the AM returns to the Atlantic through the thermohaline loop in the form of overflow. The remaining 30 % of the Atlantic inflow enters the estuarine loop where it joins the BS-inflow and the remainder of the Freshwater input. With the numbers in Figure 12, Atlantic inflow supplies around 70 % of the total surface outflow and BS-inflow somewhat more than 25 %, but these numbers are of course

sensitive to the uncertainties involved.

## 4.6 Long-term variations of the AM-exchanges

The exchanges between the AM and the rest of the world oceans, the AM-exchanges, are an integral part of the AMOC. With a total volume transport close to 6 Sv (Figure 12), the overflows contribute the densest third to the production of

NADW. The overflow contribution to the NADW is furthermore augmented by the waters entrained downstream of the Greenland-Scotland Ridge (e.g. Fogelqvist et al., 2003) and probably exceeds the additional contribution from convection in the Labrador and Irminger seas (Dickson and Brown, 1994; Hansen et al., 2004).

With that in mind, the projected weakening of the AMOC (Collins et al., 2013) might well be expected to affect the AM-exchanges, but a closer scrutiny of the different climate models demonstrates huge differences in the projections for that

part of the AMOC that involves the AM (Sgubin et al., 2017). For most of the world, it may not be important which source for the AMOC weakens, but that is not the case for the regions affected by the pole-ward heat transport associated with the upper limb of the AMOC. Thus, conditions in the AM are critically dependent on the heat imported by the Atlantic inflows (Skagseth and Mork, 2012; Mork et al., 2014; Årthun et al., 2012; Onarheim et al., 2014; Årthun et al., 2017; Utne et al., 2012). The effect of the oceanic heat transport on Arctic sea ice (Zhang, 2015), and vice versa (Bitz et al., 2006), is also

speculated to feed back on mid-latitude weather systems, currently a research topic of high interest (e.g. www.blue-action.eu).

It is therefore highly relevant to ask, whether our data indicate any weakening over their observational periods, which exceed two decades for the longest observed branches. The brief answer to this question is no. On the contrary, the only significant trend found is the Pacific inflow, which showed an increasing (not weakening) trend (Table 2), while the Atlantic inflow as well as the two dominant overflow branches remained stable (Figure 10).

A priori, this result may seem to be in conflict with reports of AMOC weakening 2004-2012 at 26 °N (Smeed et al., 2014) especially since they found the weakening to be due to a slowing of the southward flow of "lower NADW below 3000 m" by 7% per year. Later measurements indicate that the North Atlantic Ocean went into a state of reduced overturning during the period 2008-2012 with a 30 % reduction of lower NADW between the periods 2004-2008 and 2008-2017 (Smeed et al., 2018).



Generally (e.g. Orsi et al., 2001), lower NADW has been considered to be fed from overflows and entrained waters. One might therefore expect to see this reported weakening reflected in our data.

Instead, the two main overflow branches in our data set indicate no significant weakening during this period (Figure 10b and thick blue line in Figure 11) and this result is strengthened by the behaviour of the total Atlantic inflow (thick red line in Figure 11), since the overflow and the Atlantic inflow must be strongly coupled through the thermohaline loop according to Figure 12. Our results indicate that any weakening of the AMOC during the last two decades cannot have been caused by weakened overflow or reduced overturning in the AM.

For the estuarine loop, increases have been reported for the BS-inflow (Woodgate et al., 2018) as well as the Freshwater input (e.g. Haine et al., 2015). These increases are, however, small compared to the total surface outflow. Thus, the overall picture for the AM-exchanges is one of stability. Perhaps, slight strengthening of both circulation loops, but certainly no weakening.

It should be emphasized that the observed stability of volume transports does not imply that water mass properties also have remained stable during the last two decades. Rather, temperature and salinity have varied considerably for both the Pacific and the Atlantic inflows, although overall trends have been small (Woodgate et al., 2018; González-Pola et al., 2018). More persistent changes have been observed for the densest overflow branch, the FB-overflow, which has warmed consistently since around 2002, although density has remained stable due to concurrent salinity increase (Hansen et al., 2016).

Our finding that the AM-exchanges have been stable in terms of volume transport during a period when many other components of the global climate system have changed is reassuring, but the possibility of future change remains (Sgubin et al., 2017). Continued increase of freshwater supply to the AM may act to destabilize the exchanges and so may changes in the oceanic salt transport into the AM. The coupling between the Atlantic inflow and the overflow (Figure 12) may be seen as a feedback mechanism (Stommel, 1961), which makes the thermohaline loop sensitive to the salinity of the Atlantic inflow. In this connection, the dramatic freshening of the Atlantic inflows since ~2010 (González-Pola et al., 2018) is worrisome. This emphasizes the need to maintain and ideally expand the monitoring system.

## 5 Conclusions and recommendations

Although the time series for many of the exchange branches in our data set have large gaps and are based on a non-contiguous observational system, we find that they do present a consistent picture of the total AM-exchanges. The most complete coverage is for the AM-import, consisting of the combined oceanic inflows and the Freshwater input (defined here as riverine, surface net precipitation, and glacier run off). On average, the AM-import is found to total 9.1±0.7 Sv with a fairly consistent seasonal variation that has an amplitude close to 1 Sv and maximum import around October.

It must be kept in mind, however, that these numbers may be somewhat dependent on the locations of the sections where the oceanic inflows are monitored. This is especially the case for the inflows over the Greenland-Scotland Ridge where the inflowing Atlantic water has to be distinguished from other water masses flowing through the monitoring sections. This and the fact that import and export branches through the same passages across the ridge mainly



are monitored on different sections imply that we should not expect a perfect balance between observed AM-import and AM-export.

In spite of that, our data give a good balance between average import and export with only 0.4 Sv more water being exported than imported on the average, which is well below the combined quoted uncertainties. More problematic is the imbalance in the seasonal variations indicated by our data. This may well, however, be caused by our lack of simultaneous coverage of all the export branches, especially our very limited data set for the surface outflow through the Denmark Strait.

We therefore argue that the exchange branches that have been monitored for a long time most likely do give a good representation of the long-term variations. These are the five oceanic inflow branches and the two main overflow branches and none of them weakened. Indeed the only significant trend is in the Bering Strait inflow, which shows statistically significant increase. Thus, the AM-exchanges as a whole are not likely to have weakened during the two decades from the mid-1990s to the mid-2010s. This includes the total overflow from the AM.

Certainly, the combined transport of the two main overflow branches did not weaken and they account for almost 90 % of the total overflow. Around 70 % of the Atlantic inflow is converted into overflow and the observed stability of the total Atlantic inflow further indicates that the thermohaline loop of the AM remained stable during our observational period. Although the overflow is a key component of the AMOC, any weakening of the AMOC during this period cannot have been caused by weakened overflow or weakened overturning in the AM.

In the global climate system, the AM-exchanges play several key roles. The overflows feed the AMOC whereas the surface outflows may interact with the subpolar dense water production also feeding the lower limb of the AMOC. The pole-ward heat transport of the oceanic inflows affects local climate, fish stocks, and sea ice cover with possible repercussions on mid-latitude weather systems.

With all this in mind, our finding that the exchanges have not weakened during the last two decades of global change is reassuring, but it is no guarantee of future stability. Atmospheric warming and increased freshwater supply have the potential to affect the stability of the thermohaline loop, as do changes in the oceanic inflows such as the recent dramatic freshening of the Atlantic inflows. Potential climate-induced changes in wind regimes may similarly affect especially the estuarine loop.

We recommend that more effort is put into quantifying the exchange branches that up to now have not been adequately observed. These are especially the surface outflow through the Denmark Strait, the overflow across the Iceland-Faroe Ridge, and the inflow over the Scottish shelf. With better coverage of these branches, we believe that firmer conclusions could have been reached. For most impact, the branches have to be monitored simultaneously, however, Even a one-time effort with all exchange branches monitored over a year would help substantially.

We therefore strongly recommend that all possible efforts are made to maintain the established monitoring systems. These systems are demanding in manpower and continued funding and short-term scientific discoveries are not always guaranteed, but they are the safest way to stay alert against possible future changes since it is not yet clear where and how a disruption of the AM-exchange systems will first be manifested or which indices may serve as early warning indicators.



**Data availability**

The observational data used in this study are available online at www.zenodo.org, www.envofar.fo, www.bodc.ac.uk, http://iop.apl.washington.edu/data.html and http://psc.apl.washington.edu/HLD/Bstrait/bstrait.html.

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

**Table 1** Observational characteristics of each AM-exchange branch. The full period of observations is listed with the number of months observed (Months) and the number of missing months (Gaps). The uncertainties of the average values are based on the information in Sect. 2. The average transport values (Avg.) are positive into the AM and negative out of the AM. Std is the standard deviation of the monthly averages. References to the sources for the data are listed for each branch in Sect. 2.

| Branch Full name | Branch Abbrev. | Period yyyy/mm-yyyy/mm | Months | Gaps | Avg. Sv | Std. Sv |
|---|---|---|---|---|---|---|
| **Inflows:** | | | | | | |
| Denmark Strait Atlantic | DS-inflow | 1994/10-2015/12 | 250 | 5 | 0.9±0.1 | 0.3 |
| Iceland-Faroe Atlantic | IF-inflow | 1993/01-2015/12 | 276 | 0 | 3.8±0.5 | 0.6 |
| Faroe-Shetland Atlantic | FS-inflow | 1993/01-2015/12 | 276 | 0 | 2.7±0.5 | 1.1 |
| European Shelf Atlantic | ES-inflow | 1993/01-2015/12 | 276 | 0 | 0.6±0.2 | 0.3 |
| Bering Strait Pacific | BS-inflow | 1997/08-2013/12 | 197 | 0 | 0.9±0.1 | 0.4 |
| **Overflows:** | | | | | | |
| Denmark Strait | DS-overflow | 1996/05-2015/12 | 218 | 18 | -3.2±0.5 | 0.4 |
| Iceland Faroe Ridge | IF-overflow | n.a. | | | -0.4±0.3 | |
| Faroe Bank Channel | FB-overflow | 1995/12-2015/12 | 206 | 35 | -2.0±0.3 | 0.3 |
| Wyville Thomson Ridge | WT-overflow | 2006/05-2013/05 | 61 | 24 | -0.2±0.1 | 0.1 |
| **Surface outflows:** | | | | | | |
| Canadian Archipelago | CA-outflow | 2004/10-2010/09 | 72 | 0 | -1.7±0.2 | 0.7 |
| Denmark Strait | DS-outflow | 2011/09-2012/07 | 11 | 0 | -2.0±0.5 | 0.5 |
| **Runoff and precipitation:** | | | | | | |
| Freshwater input | Freshwater | n.a. | | | 0.2 | |

**Table 2.** Linear trends of annual averages of the five inflow branches individually and summed and of the DS-overflow. Only years with complete coverage (no months missing) are included and the number of years is listed. The trend is represented by its value ± its 95% confidence interval. Branches with trends that significant at the 95% level are marked in bold. The last column lists relative trends determined by dividing the trends with the




average transports from Table 1.

| Branch | Period | Years | Trend (Sv yr$^{-1}$) | Rel. tr. (yr$^{-1}$) |
|---|---|---|---|---|
| DS-inflow | 1997-2015 | 18 | 0.004±0.011 | 0.4 % |
| IF-inflow | 1993-2015 | 23 | 0.012±0.013 | 0.3 % |
| FS-inflow | 1993-2015 | 23 | -0.006±0.024 | -0.2 % |
| ES-inflow | 1993-2015 | 23 | 0.003±0.005 | 0.5 % |
| BS-inflow | 1998-2013 | 16 | **0.016±0.014** | **1.8 %** |
| All inflows | 1998-2013 | 15 | 0.040±0.046 | 0.4 % |
| DS-overflow | 1997-2015 | 14 | -0.007±0.015 | -0.2 % |





**Figure 1: The Arctic Mediterranean (roughly represented by the oceanic areas within the yellow curve) and its exchanges with the rest of the World Ocean. Land areas are black. Ocean areas shallower than 1000 m are light grey. Red arrows indicate inflow branches. Dark blue arrows indicate overflow branches. Green arrows indicate surface outflow branches. Labels for arrows refer to Sect. 2 with numbers indicating average volume transport in Sv (1 Sv = 106 m3 s-1) based on Table 1.**

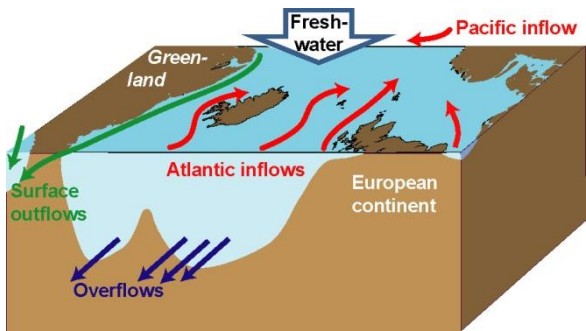

**Figure 2: In this study, the oceanic flows into and out of the AM are grouped into three categories: inflows, overflows, and (surface) outflows. In addition, rivers, Greenland meltwater discharge, and net ocean surface precipitation supply freshwater to the AM.**

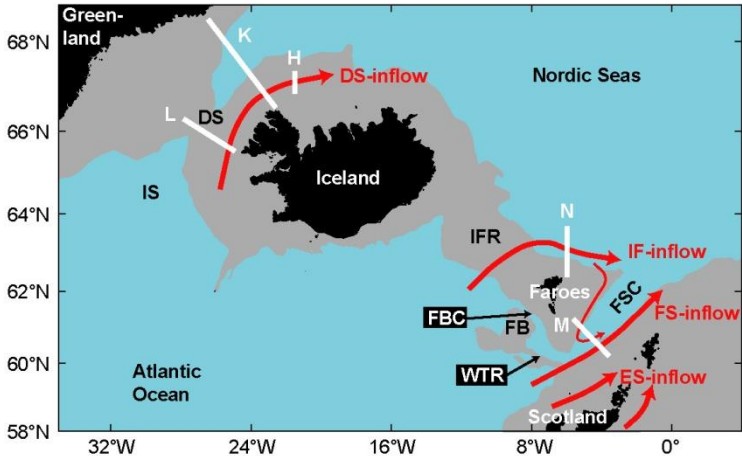

**Figure 3: The Greenland-Scotland Ridge. Grey areas are shallower than 750 m. Red arrows show schematic flow patterns of the four Atlantic inflow branches. Thick white lines indicate monitoring sections with labels referred to in the text (Section 2.1). Topographic features indicated are Denmark Strait (DS), Irminger Sea (IS), Iceland-Faroe Ridge (IFR), Faroe Bank (FB), Faroe Bank Channel (FBC), Faroe-Shetland Channel (FSC), and Wyville Thomson Ridge (WTR).**





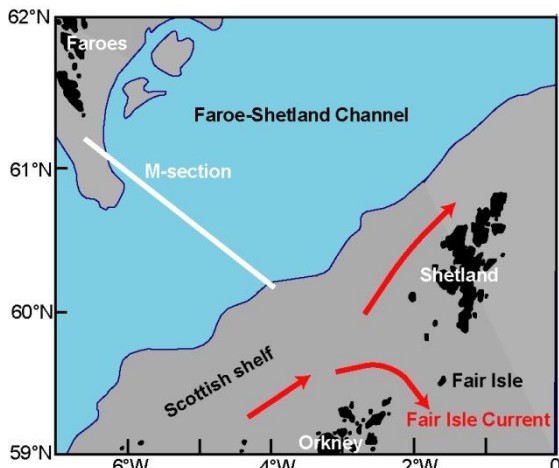

**Figure 4: The northern component of the ES-inflow (Section 2.1.4) shown by red arrows. Grey areas are shallower than 200 m. White line indicates the M-section, on which the FS-inflow (Section 2.1.3) is monitored.**

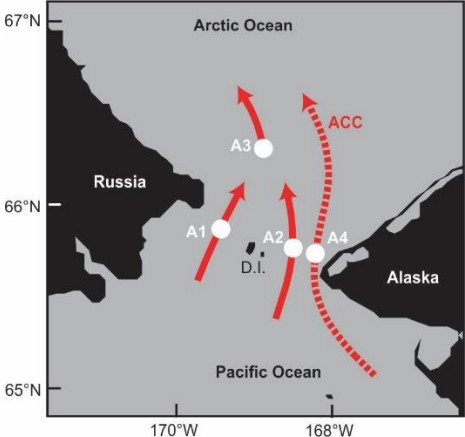

5 **Figure 5: The Bering Strait has two channels separated by the Diomede Islands (D.I.). Red arrows indicate annual mean flow paths. White circles mark mooring positions (A1, A2, A3, A4). Dashed arrow marks the Alaskan Coastal Current (ACC), present seasonally. Grey areas are shallower than 100 m.**



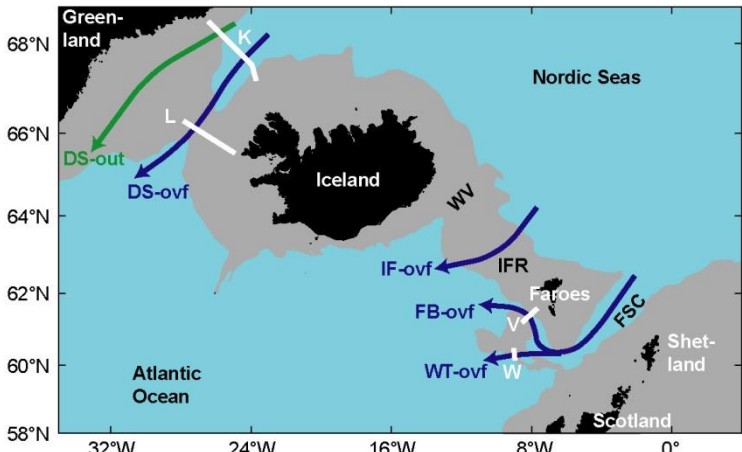

**Figure 6:** Overflow and surface outflow branches across the Greenland-Scotland Ridge. Grey area is shallower than 750 m. Arrows indicate schematic flow patterns of the four overflow branches (dark blue, discussed in section 2.2) and the one surface outflow across the ridge (green, discussed in Section 2.3). Thick white lines indicate monitoring sections with labels referred to in the text.
5   Topographic features indicated are Iceland-Faroe Ridge (IFR), Faroe-Shetland Channel (FSC), Western Valley (WV).

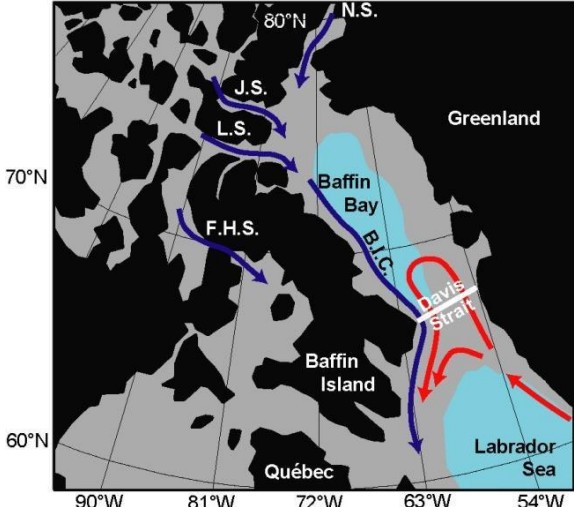

**Figure 7:** Outflow from the Arctic Ocean through the Canadian Archipelago. Flows through Nares Strait (N.S.), Jones Sound (J.S.), Lancaster Sound (L.S.), and Fury and Hecla Strait (F.H.S.) as well as the Baffin Island Current (B.I.C.) are indicated by blue arrows. The thick white line indicates the Davis Strait monitoring section. Red arrows indicate water flowing northwards through the section
10  before recirculating, joining, and partly mixing with the Arctic Ocean outflow and exiting south again. Grey areas on the map are shallower than 1000 m.



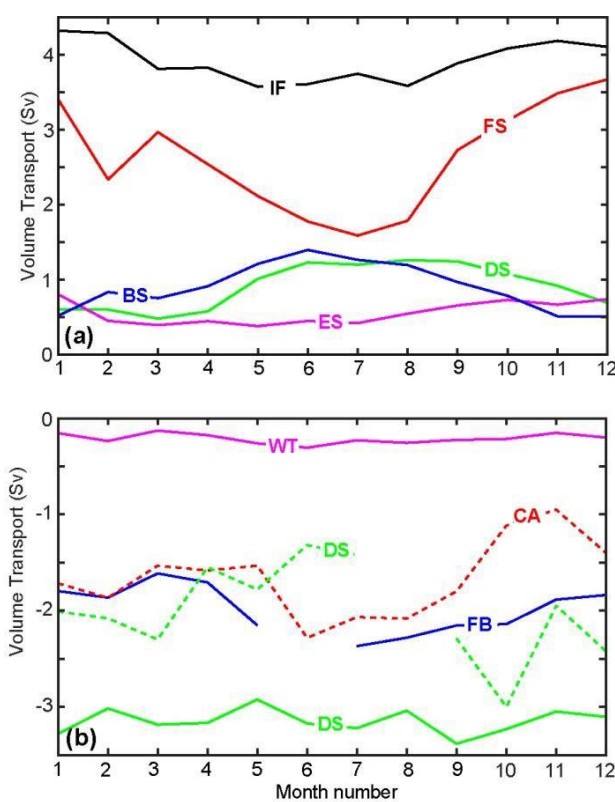

**Figure 8: Seasonal variation of five inflow branches (a), three overflow branches (continuous lines in b), and two surface outflow branches (dashed lines in b). All the lines are based on observations taken between October 2004 and September 2010 except for the DS-outflow (dashed green line in b), which is based on the September 2011 to July 2012 period with inshore values from 2013-2014**
5   **(Sect. 2.3.2). We have no seasonal information for the IF-overflow and so it is not included in this plot. See Table 1 for abbreviations.**

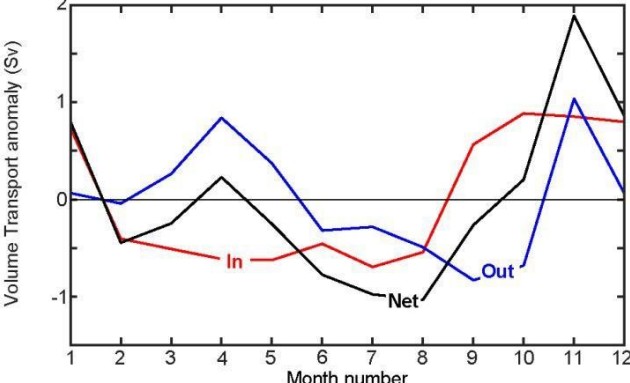

**Figure 9: Seasonal anomalies of the combined inflow branches (red) and the combined overflow and surface outflow branches (blue) for the same periods as in Figure 8, where missing months have been interpolated. The black curve is the sum of the other two curves and represents the anomaly of the Net import when the IF-overflow (order 0.4 Sv in the annual mean) and Freshwater (order 0.2 Sv**
10   **in the annual mean) input are neglected.**




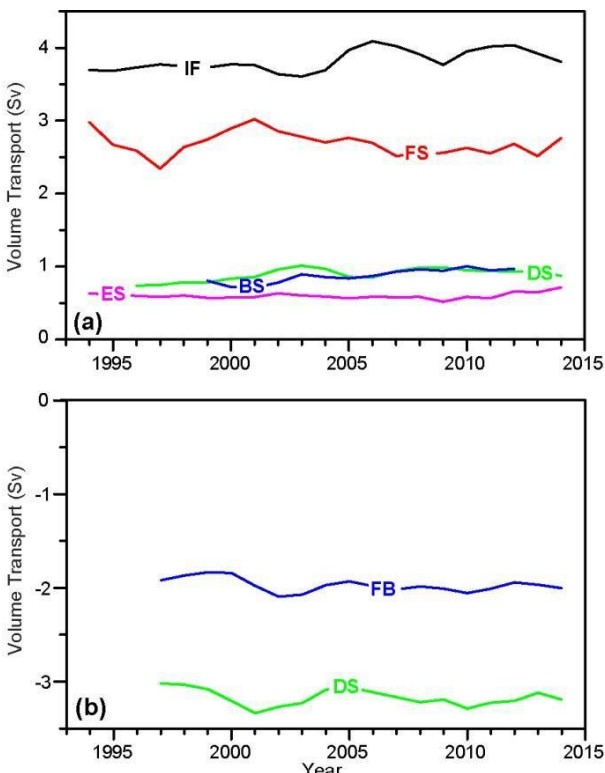

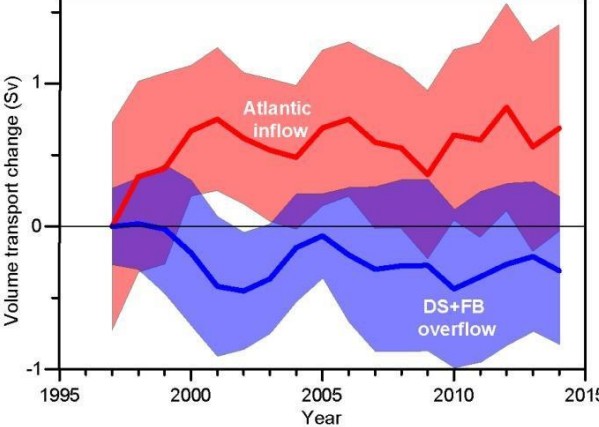

**Figure 10: Low-passed (3-year running mean) volume transport of the five inflow branches (a) and the two main overflow branches (b). The value for each year is the average of the values for all observed months of the year, the preceding year, and the following year. To minimize bias from missing months, the values have been de-seasoned before averaging (Sect. S2 in the supplementary document).**

**Figure 11: Low-passed (3-year running mean) volume transport change (from the value in 1997) of the sum of the four Atlantic inflow branches (thick red line) and the sum of the two main overflow branches (thick blue line). The value for each year is the average of the de-seasoned values for all observed months of the year, the preceding year, and the following year. The coloured areas represent the 95% confidence interval (Sect. S2).**





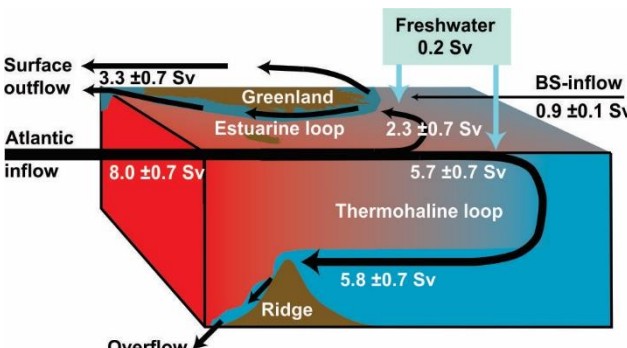

**Figure 12: Overall budget for the AM-exchanges where the circulation within the AM is simplified into two loops: a thermohaline loop converting Atlantic inflow and freshwater into overflow, and an estuarine loop converting all three types of AM-import into surface outflow.**