# Peer review of "Arctic Mediterranean Exchanges: A consistent volume budget and trends in transports from two decades of observations"

_Ocean Science, 2018_

## Referee Comment (RC1) · W.-J. von Appen (Referee) · 20 Nov 2018

The manuscript is the conclusion of a multi-year effort to bring together information from the exchanges between the Arctic Mediterranean and the rest of the world's oceans. This is an important update of work that was started within the framework of the Arctic-Subarctic-Ocean-Fluxes and maintained since. The implications of these exchanges for the regional and global oceanography are clear and important. The manuscript is written in a straightforward way, of good presentation quality, and therefore easily readable. As such I can recommend publication of this manuscript in Ocean Science after a minor revision.

[Figure]

Major comments:

One thing that is probably an explicit choice, but does not always work, is that the authors do not consider any information provided about these exchanges by models. I have remarked in the minor comments below where at least a few sentences could be added.

I was a bit surprised that this recent paper which also brings together observational information from most of the same gateways discussed was not mentioned: Bringedahl JClim doi:10.1175/JCLI-D-17-0889.1 At least a reference to it and how those seasonal cycles and long-term time series agree and/or differ seems warranted.

There are many places in the manuscript (e.g. p1l31, p4l3, p16l15/23/29/30) where subscripts and superscripts were not converted correctly into the typeset version.

Minor comments line by line:

p1l31 9.1+-0.7Sv What does the "+-" refer to? Is it the standard deviation? Of what? Please specify.

p1l38 "At the 95% confidence level"

p2l29 and p9l30 "en route" instead of "on route"

p3l5 Somehow the grouping should be different. CAA should be separate from the combined outflow route of FS/DS.

p4l7 "without yielding any information"

p5l13 Many months have 31 days whereas February has 28 days in most years. Has this difference been taken into account? Or in order to arrive at an annual value, did you simply take the sum of (January average + February average + March average + . . .) divided by 12?

p5l17 "but is deeper" Should it not rather be "shallower" or do you need a different

conjunction than "but"?

p7l1 "it seems clear" Why does it seem clear? To me it is not clear at this point.

p10l10 Are those 0.2Sv accounted for in the surface outflows?

p11l9 Is "Canadian Arctic Archipelago" not a more common term than "Canadian Archipelago"?

p11l18 "mooring array north of the sill"

p14l22 "serial correlation" It is not clear exactly what is meant by that term. Please elaborate in 1-2 sentences.

p14l24 Consider "The exchanges between the AM and the Atlantic are therefore characterized by stability rather than change—at least over the observed period."

p16 While it is in principal mathematically correct to define tauH and tauQ and relate them to each other (equation 9), in my point of view, this is needlessly confusing. The more straightforward way would be to substitute cos by sin in equation 2 and to have the same phase tauQ there.

p1730-32 What is "wanted" and "unwanted" water? Is not all of the water passing the sections water that passes the sections and therefore to be considered? Maybe I'm just confused by the terminology.

p18l5-7 Are you referring to non-linear effects of correlations between transport and water mass variability on higher frequencies than monthly (e.g. "eddy correlations")? If so, it is not clear to me why this should be random and small. Rather this could introduce a systematic (rather than random) bias whose magnitude is not clear a priori.

p19l20 This would be a good place to spend at least 1-3 sentences on what models have to say about this point. While your paper is observationally in its focus, you can at least refer to model results for hypotheses/speculation.

[Figure]

p19l34 "in Fram Strait in some years"

p21l6-7 "cannot have been caused" Also in light of your later sentence (p21l12-13) I think this statement is too strong. Given that changes in overflow properties (density in particular) can non-linearly lead to changes in the AMOC even for a constant overflow volume, you could point out that the overflow volume has not changed while you are not focussing on the other properties.

p22l29 "... simultaneously. However, even ..."

p23l2/3 Could you provide more complete links (not just the main website domain) or even more appropriately DOIs?

p30l20 "trends that are significant"

p33l7 "Grey areas ..." On the shelf this makes for a humorous statement. I presume that was intended...

Fig8/Fig10 Both of these figures do not need a panel (a) and panel (b) which then have different spacings on the y-axis. Rather you could have a single panel with the y-axis ranging from -3.5Sv to 4.5Sv. This would make a visual comparison a lot less difficult.

Fig8 In this way, visually January and December are represented as half months while the other 10 months take up more space per month. This again makes a visual assessment of what is happening more difficult than necessary. Put another way, the line connecting December to January is missing while it is present (and occupying the visuals) for the other months.

---

## Referee Comment (RC2) · Anonymous Referee #2 · 21 Jan 2019

The submitted manuscript presents a comprehensive update of the Arctic Mediterranean (AM) oceanic volume fluxes based on the long-term observational efforts in the different branches of Atlantic inflows, and surface and deep outflows (overflows). Based on available transport estimates, the authors aim to construct a physically consistent volume budget within the estimated uncertainties of discussed AM exchanges and, where possible, to evaluate the seasonal variations of volume transport in individual branches (and in the integrated Atlantic inflow). The long-term trends in the AW exchange branches are discussed in the context of the recently widely debated weakening of the AMOC with reaching the important conclusion about a lack of observational evidence thereof. The manuscript not only provides the most comprehensive

and up-to-date overview of the AM exchanges, but also addresses the limitations of existing observational arrays/activities in different inflow and outflow branches, and discusses uncertainties introduced by their imperfect coverage by long-term measurements. The submitted paper is clearly structured and divided into the sections focused on individual AM exchange branches, followed by the chapters integrating the partial results towards the overall budget and its seasonal and long-term variability. The final conclusions are well justified by presented results and discussion. The comprehensive table summarizes the details of volume transports in all addressed branches while a complex spatial structure of AM exchanges is illustrated with detailed maps, including a conceptual scheme of two circulation loops representing the whole budget. The submitted paper offers a much-needed synthesis of the current knowledge about volume exchanges between the AM and the World Ocean that have a profound impact both on regional circulation and global climate-relevant processes. The scientific content is sound, the results solid, and the manuscript is generally well written, so the suggested modifications/additions are not critical but intended to improve readability and clarify the details. Therefore, I would like to recommend the manuscript for publication in the Ocean Science journal after a minor revision.

General comments: While the manuscript provides a wealth of detailed information on available measurements in the discussed branches of AM exchanges, little attempt has been made to compare the obtained budget and variability to other existing estimates, based on numerical models, reanalyses or other observations (e.g. satellite altimetry) – or a combination thereof. Two recent publications with a similar focus but different approach would be the obvious candidates for such comparison: Bringedahl et al. (2018, Journal of Climate) for time series of volume transports and seasonal variability, and Rossby et al. (2018, J. Geophys. Res. Oceans) for volume exchanges across the across the Greenland-Iceland-Faroe-Scotland Ridge. How do volume transports and their seasonal variations presented in the manuscript compare to the estimates obtained farther south, at the Ovide or OSNAP sections (e.g. Daniault et al., 2016, Prog. Oceanogr., or Gary et al., 2018, J. Geophys. Res. Oceans) or along the other lines,

closing the passage between Greenland and Scotland (e.g. Chafik et al., 2014, J. Geophys. Res. Oceans). It would be very interesting to consider the presented fluxes in a wider context. Another interesting question would be how well the proposed budget concur with the constraints for exchanges in the Arctic Mediterranean as elaborated by Rudels (2010, Tellus A). While the simplified concept of a double estuary with two circulatory loops serves as a good representation of the overall budget, I would appreciate a more thorough discussion of how much of the Atlantic inflow, modified along different pathways in the AM and returning to the North Atlantic, is not accounted for by the measured combination of surface outflows on both sides of Greenland and deep overflows in different branches.

Specific comments: Page 1 line 23: Should be '. . .is modified within the AM.'

Page 1 line 26: '. . .heat, salt and other substances. . .', '. . .are important for conditions in the AM'

These statements sound a little vague, please be more precise about 'other substances' and 'conditions'.

Page 1 line 31: Superscripts are not correctly typeset here (and also in many following instances in the text).

Page 1 line 31-32: '. . .has a seasonal variation of amplitude close to 1 Sv'

I would rather suggest 'has the amplitude of the seasonal variation close to 1 Sv'

Page 1 line 33: 'The overflow is mainly produced. . .'

I suggest 'The overflow water is mainly produced. . .'

Page 1 line 35: '. . .is fed from the Pacific inflow and freshwater'

I suggest adding the origin of freshwater in this sentence.

Page 1 line 35: '. . .is ∼2/3rds from modified Atlantic water.'

I would suggest '. . .is ∼2/3rds of modified Atlantic water'.

Page 1 line 38: 'At the 95% level. . .'

It should be 'At the 95% confidence level. . .'

Page 2 line 16: '. . .transporting heat, salt and other substances.'

As above – what other substances?

Page 2 line 27: '. . .as "overflow" waters.'

Why to use the quotation marks here? Overflow water is well-accepted name for this water mass.

Page 2 line 29: '. . .entrain on route. . .'

Either 'en route' or 'on the way'.

Page 3 lines 4-6: 'The inflowing water from the Atlantic. . .'

What about the part of Atlantic water that recirculates along different loops in the Nordic Seas and Arctic Ocean and does not return as 'cold and fresh surface outflow' but rather occupies the subsurface and intermediate layers when flowing to the south?

Page 3 line 22: '. . .expected to be qualitatively different. . .'

This statement sounds a little peculiar. If it was meant that the budget has different components (different flow branches) then it is quantitatively different. On the other hand, the volume (mass) budget should be closed both for the AM and for the Arctic Ocean thus it cannot be 'qualitatively different'. I would suggest reformulating this sentence.

Page 4 line 7: 'without any yielding any information. . .'

One 'any' too many. . . (without yielding any information).

Page 4 line 9: 'the variability in physical aspects. . .'

This sounds somehow cryptic. What are the differences between individual branches that make them difficult to be described in a consistent manner?

Page 4 line 17: '. . .for historical and logistical reasons. . .'

What are 'logistical' reasons? Do you mean the distribution/locations of observations or the structure of paper?

Page 4 lines 25-26: 'Over the deepest part. . . . . . towards the Irminger Sea'.

This sentence does not belong here as it describes the outflow (DSOW flowing towards the Irminger Sea), not inflow. The same refers to the previous sentence where the surface outflow in the EGC is described. I would suggest keeping the description of inflows and outflows separate.

Page 4 line 27 and Fig. 3: Why are these two branches not shown on Figure 3?

Page 5 line 34, page 5 line 1: This sentence is difficult to follow (in particular 'are used' at its end), please reformulate.

Page 5 lines 25-27: Does the Faroe Current as measured at the section N include the entire flow of AW passing between Iceland and Faroe or is there any part that passes northward beyond the section and is not accounted for?

Page 6 lines 5-6: '. . .a significant fraction originally crossed. . . . . .bifurcated into the FSC. . .'

The verb tenses are strange here. I would suggest '. . .a significant fraction that originally crossed the ridge. . . . . .enters the Faroe Current and bifurcates into the FSC. . .'.

Page 8 line 27: '. . . on the Greenland shelf region. . .'

Either 'on the Greenland shelf' or 'in the Greenland shelf region. . .'

Page 9 line 19: '. . .uncertainty (estimated from their figures). . .'

How was the uncertainty estimated from the figures?

Page 10 line 9: '. . .the "kinematic overflow", has an average volume transport of. . .'

What is a difference overflow and 'kinematic overflow'? Is the latter one defined not by density range but some other criteria?

Page 10 line 30: 'The definition of FSCBW is denser than our criterion. . .'

Is the assumed source FSCBW denser that the criterion for overflow water or the mixture between FSCBW and AW? The criterion used by Johnson et al. (2017) is on the other hand less dense therefore 0.3 Sv may by overestimated.

Page 11 line 9 and following: 'Canadian Archipelago. . .'

The commonly accepted name is the Canadian Arctic Archipelago (CAA).

Page 11 lines 16-17: '. . .carries inputs from the integrated CA outflow as well as northward inflow. . .'

Perhaps it could be helpful here to mention a different origin (and characteristics) of water masses in the integrated CAA outflow and in the recirculating flow from the West Greenland shelf and slope. A more precise way to describe the outflow from the Davis Strait would be 'the integrated CAA throughflow and modified AW recirculating from the West Greenland Current'.

Page 13 lines 22-23: 'The sum of the transport values. . . . . .did not, however, differ substantially from the sum based on the full periods'

Even if the sum of transport values did not differ substantially, it would be helpful to be able to compare the 6-year averages of volume transport for individual branches with those based on the full periods. Perhaps one more column could be included in Table 1 to show transports averaged for the reference (overlapping) period, especially when taking into account that monthly averages over this period are later employed to analyze the seasonal variations.

Page 14 line 22: '. . .without taking serial correlations into account. . .'
Please explain more precisely how would accounting for autocorrelation increase the confidence intervals for calculation of trends in volume transports.

Page 17 line 9: '. . . which supports the value of 5 cm as a maximum in the AM as a whole.'

Could you elaborate more precisely how is the maximum value of 5 cm for the whole AM obtained from the sea level variations south of 82°N.

Page 17 line 13: '. . .is not very consistent with this.'

I would not call it 'not very consistent' but not consistent at all since there is a difference on the order of magnitude between the seasonal amplitude estimated from the sea level variation and seasonal amplitude based on volume transport measurements.

Page 18 lines 1-8: I would be more careful about downplaying the uncertainties related to different criteria used to distinguishing water masses in inflows and out-flow/overflows. The relationship between flow (transport) and hydrographic characteristics at the section is not necessarily linear and it is unclear to me why possible differences should result in systematic biases, not the random errors.

Page 18 line 8: '. . .budget of the AM and data quality'

The phrase 'Data quality' does not reflect the core of the problem as the data quality is the most likely acceptable for this kind of large-scale estimates. The problem is in too sparse measurements, so I would rather suggest '. . .budget of the AM and gaps in the observational coverage'.

Page 18 lines 19-20: '. . .in the form of an unknown bias rather than a randomly varying error. . .'

As mentioned above, I am not convinced that this is necessarily the case.

Page 19 line 5: 'Combining the uncertainties. . . . . .quadratically, as commonly done. . .'

'Combining quadratically' sounds a little peculiar, please reformulate into the assumption about error propagation. Why should it also be 'a conservative estimate' of the overall uncertainty?

Page 19 lines 18-19: '...most of this seasonality would have to come from the DS-outflow, i.e. the estuarine loop...'

The meaning of the estuarine and thermohaline loops should be introduced before discussing their roles in the seasonal variability.

Page 19 lines 21-22: '...if the monitoring of the various import and export branches in the Greenland-Scotland region had been better coordinate with identical monitoring sections for import and export branches.'

The meaning of this sentence is entirely incomprehensible to me. Do you mean coordination in time (concurrent monitoring)?

Page 19 line 34: Should be 'in some years...'

Page 20 line 3: 'in a mixing rate of 99:1...'

Where does this estimate of mixing ratio come from (it is not clear from the given salinities for AW and OW)?

Page 20 lines 15-17: Could you provide at least rough estimates for the additional contributions from the entrainment and convection?

Page 21 lines 10-11: 'Perhaps, slight strengthening of both circulation loops but certainly no weakening'.

This sounds as a speculative statement. Please elaborate more precisely and formulate as a full sentence.

Page 22 lines 7-8: 'We argue that the exchange branches that have been monitored for a long time most likely do give a good representation of the long-term variations'

The sentence that longer observations provide better estimates of long-term variations is a truism. I would suggest using more precise formulation here.

Page 22 lines 30: '... a one-time effort with all exchange branches monitored over a year would help substantially.'

While this is the most likely true, a concrete argument how would it help would be more convincing (e.g. elucidating relations between transports in different branches, lower uncertainties, etc.)

Page 32 Figure 3 and page 33 Figure 4: My suggestion is to slightly enlarge Figure 3 towards the south and incorporate the arrows showing the ES inflow into it. Figure 4 is in my opinion superfluous.

Page 34 Figure 7. Why are the abbreviations of currents' names with dots (periods) on this figure (and in its caption) and without periods on other figures.

Page 35 Figure 8 and page 36 Figure 10: I would suggest combining panels (a) and (b) into one plot for each of these figures and, in the first place, using one Y-scale for inflows and outflows to be able to compare their variations.

---

## Author Comment (AC1) · 11 Mar 2019

Reply to Referee # 1 (W.-J. von Appen)

Please find our answer to Referee #1 below. (A more Reader friendly PDF file is included as supplement).

Major comments: One thing that is probably an explicit choice, but does not always work, is that the authors do not consider any information provided about these exchanges by models. I have remarked in the minor comments below where at least a few sentences could be added.

[Figure]
* * *
ANSWER Yes it is explicit Choice. We agree that compering our direct volume trans-
port observations with information provided by models would add value to this paper,
but we have deliberately chosen to give a conscientious description and analyses of
our observations. However, in future works we will compare our observations with nu-
merical models other observations to discuss our results in wider context. We have
added a sentence in sect. 4.4: "…. but will have to await future observational efforts
for confirmation. Meanwhile our time series will be combined with results from numeri-
cal models, reanalyses (Bringedal et al., 2018) and observations using other methods
(Rossby et al., 2018)"
* * *
I was a bit surprised that this recent paper which also brings together observational
information from most of the same gateways discussed was not mentioned: Bringedahl
JClim doi:10.1175/JCLI-D-17-0889.1 At least a reference to it and how those seasonal
cycles and long-term time series agree and/or differ seems warranted.
* * *
ANSWER A reference to Bringedal et al. is added in sect. 4.4
* * *
There are many places in the manuscript (e.g. p1l31, p4l3, p16l15/23/29/30) where
subscripts and superscripts were not converted correctly into the typeset version.
* * *
ANSWER corrected
* * *
Minor comments line by line:

—————————————

p1l31 9.1+-0.7Sv What does the "+-" refer to? Is it the standard deviation? Of what? Please specify.

—————————————————

ANSWER The sentence has been reformulated

——————————————-

p1l38 "At the 95% confidence level"

——————————————————-

ANSWER Changed accordingly

——————————————

p2l29 and p9l30 "en route" instead of "on route"

—————————————-

ANSWER Changed accordingly

—————————————————

p3l5 Somehow the grouping should be different. CAA should be separate from the combined outflow route of FS/DS.

————————————————————-

ANSWER Changed to: . . . and leaves the AM through the Canadian Archipelago and Denmark Strait and the upper western Fram Strait as cold . . .

————————————————-

p4l7 "without yielding any information"

—————————————————-

ANSWER Changed accordingly

———————————————

p5l13 Many months have 31 days whereas February has 28 days in most years. Has this difference been taken into account? Or in order to arrive at an annual value, did you simply take the sum of (January average + February average + March average +: : :) divided by 12?

——————————————— ANSWER We have added a clarifying sentence to the beginning of Sect. 3

—————————————-

p5l17 "but is deeper" Should it not rather be "shallower" or do you need a different conjunction than "but"?

————————————

ANSWER "but" has been changed to "and"

—————————————————-

p7l1 "it seems clear" Why does it seem clear? To me it is not clear at this point.

——————————————

ANSWER Deleted: "it seems clear that"

———————————————

p10l10 Are those 0.2Sv accounted for in the surface outflows?

———————————————————-

ANSWER Part of this water, at least, is Atlantic water entrained into the overflow along

its path from the Faroe-Shetland Channel into the Faroe Bank Channel. This is one of the problems more generally addressed in Sect. 4
* * *
p11l9 Is "Canadian Arctic Archipelago" not a more common term than "Canadian Archipelago"?

_________________________________-

ANSWER Changed to Canadian Arctic Archipelago (CA)

_______________________________-

p11l18 "mooring array north of the sill"

______________________________-

ANSWER Changed accordingly
* * *
p14l22 "serial correlation" It is not clear exactly what is meant by that term. Please elaborate in 1-2 sentences.
* * *
ANSWER The word "autocorrelation" has been added and more text p14l24 Consider "The exchanges between the AM and the Atlantic are therefore characterized by stability rather than change—at least over the observed period."
* * *
ANSWER Changed accordingly
* * *
p16 While it is in principal mathematically correct to define tauH and tauQ and relate

them to each other (equation 9), in my point of view, this is needlessly confusing. The more straightforward way would be to substitute cos by sin in equation 2 and to have the same phase tauQ there.

———————————————————————————

ANSWER We have modified the equation to include the sin version also, but kept the original (cos) version as well, because we want to define the tauH and show that Q(t) is maximum 3 months (T/4) before H(t) (Eq. (3))

———————————————————

p1730-32 What is "wanted" and "unwanted" water? Is not all of the water passing the sections water that passes the sections and therefore to be considered? Maybe I'm just confused by the terminology.

———————————————————

ANSWER This text has been modified to clarify the meaning.

———————————————————

p18l5-7 Are you referring to non-linear effects of correlations between transport and water mass variability on higher frequencies than monthly (e.g. "eddy correlations")? If so, it is not clear to me why this should be random and small. Rather this could introduce a systematic (rather than random) bias whose magnitude is not clear a priori.

———————————————————————

ANSWER This argument has been deleted from the text here and elsewhere

———————————————————————

p19l20 This would be a good place to spend at least 1-3 sentences on what models have to say about this point. While your paper is observationally in its focus, you can at least refer to model results for hypotheses/speculation.
————————————————————————

ANSWER We have added a sentence to sect. 4.4 :

—————————————————————————-

". . .. but will have to await future observational efforts for confirmation. Meanwhile our time series will be combined with results from numerical models, reanalyses (Bringedal et al., 2018) and observations using other methods (Rossby et al., 2018)"

————————————————————

p19l34 "in Fram Strait in some years"

——————————————————————-

ANSWER Changed accordingly

————————————————————

p21l6-7 "cannot have been caused" Also in light of your later sentence (p21l12-13) I think this statement is too strong. Given that changes in overflow properties (density in particular) can non-linearly lead to changes in the AMOC even for a constant overflow volume, you could point out that the overflow volume has not changed while you are not focussing on the other properties.

————————————————————————

ANSWER The text has been modified to be more specific as suggested

————————————————————————

p22l29 ": : : simultaneously. However, even : : :"

——————————————————————————— ANSWER Changed accordingly

—————————————————————————

p23l2/3 Could you provide more complete links (not just the main website domain) or even more appropriately DOIs?

———————————————————-

ANSWER We have added more completed data links.

——————————————————

p30l20 "trends that are significant"

—————————————————-

ANSWER Changed accordingly

——————————————————-

p33l7 "Grey areas : : :" On the shelf this makes for a humorous statement. I presume that was intended: : :

—————————————

ANSWER yes

——————————————————

Fig8/Fig10 Both of these figures do not need a panel (a) and panel (b) which then have different spacings on the y-axis. Rather you could have a single panel with the y-axis ranging from -3.5Sv to 4.5Sv. This would make a visual comparison a lot less difficult. ANSWER This has been done (new figures 7 and 9)

———————————————————————

Fig8 In this way, visually January and December are represented as half months while the other 10 months take up more space per month. This again makes a visual assessment of what is happening more difficult than necessary. Put another way, the line connecting December to January is missing while it is present (and occupying the

visuals) for the other months.
* * *
ANSWER The figure (now Figure 7) has been modified accordingly and new Figure 8
has also been modified in this way

Please also note the supplement to this comment:
https://www.ocean-sci-discuss.net/os-2018-114/os-2018-114-AC1-supplement.pdf
* * *

---

## Author Comment (AC2) · 11 Mar 2019

Reply to Referee # 2

Please find our answer to Referee #2 below (a more reader friendly PDF file is included as supplement).

General comments: While the manuscript provides a wealth of detailed information on available measurements in the discussed branches of AM exchanges, little attempt has been made to compare the obtained budget and variability to other existing estimates, based on numerical models, reanalyses or other observations (e.g. satellite altimetry)

– or a combination thereof. Two recent publications with a similar focus but different approach would be the obvious candidates for such comparison: Bringedahl et al. (2018, Journal of Climate) for time series of volume transports and seasonal variability, and Rossby et al. (2018, J. Geophys. Res. Oceans) for volume exchanges across the across the Greenland-Iceland-Faroe-Scotland Ridge. How do volume transports and their seasonal variations presented in the manuscript compare to the estimates obtained farther south, at the Ovide or OSNAP sections (e.g. Daniault et al., 2016, Prog. Oceanogr., or Gary et al., 2018, J. Geophys. Res. Oceans) or along the other lines, closing the passage between Greenland and Scotland (e.g. Chafik et al., 2014, J. Geophys. Res. Oceans). It would be very interesting to consider the presented fluxes in a wider context. Another interesting question would be how well the proposed budget concur with the constraints for exchanges in the Arctic Mediterranean as elaborated by Rudels (2010, Tellus A). While the simplified concept of a double estuary with two circulatory loops serves as a good representation of the overall budget, I would appreciate a more thorough discussion of how much of the Atlantic inflow, modified along different pathways in the AM and returning to the North Atlantic, is not accounted for by the measured combination of surface outflows on both sides of Greenland and deep overflows in different branches.

––––––

ANSWER We fully agree that compering our direct volume transport observations with estimates build on other methods would add value to this paper, but here we have deliberately chosen to give a conscientious description and analyses of our observations. However, in future works we will compare our observations with numerical models other observations to discuss our results in wider context. We have added a sentence in sect. 4.4: ". . .. but will have to await future observational efforts for confirmation. Meanwhile our time series will be combined with results from numerical models, reanalyses (Bringedal et al., 2018) and observations using other methods (Rossby et al., 2018)"

————————————

Specific comments:

—————————————-

Page 1 line 23: Should be '. . .is modified within the AM.'

—————————————

ANSWER Changed accordingly

———————————

Page 1 line 26: '. . .heat, salt and other substances. . .', '. . .are important for conditions in the AM' These statements sound a little vague, please be more precise about 'other substances' and 'conditions'.

————————————

ANSWER Changed to" . . . heat and salt."

————————————

Page 1 line 31: Superscripts are not correctly typeset here (and also in many following instances in the text).

————————————-

ANSWER Changes made

———————————————

Page 1 line 31-32: '. . .has a seasonal variation of amplitude close to 1 Sv' I would rather suggest 'has the amplitude of the seasonal variation close to 1 Sv'

—————————————-

ANSWER Changed accordingly

\_\_\_\_\_\_\_\_\_\_\_\_\_\_\_\_\_\_\_\_-

Page 1 line 33: 'The overflow is mainly produced. . .' I suggest 'The overflow water is mainly produced. . .'

\_\_\_\_\_\_\_\_\_\_\_\_\_\_\_\_\_\_-

ANSWER Changed accordingly

\_\_\_\_\_\_\_\_\_\_\_\_\_\_\_\_\_\_

Page 1 line 35: '. . .is fed from the Pacific inflow and freshwater' I suggest adding the origin of freshwater in this sentence.

\_\_\_\_\_\_\_\_\_\_\_\_\_\_\_

ANSWER Added: (runoff and precipitation)

\_\_\_\_\_\_\_\_\_\_\_\_\_\_\_\_\_\_\_-

Page 1 line 35: '. . .is _2/3rds from modified Atlantic water.' I would suggest '. . .is _2/3rds of modified Atlantic water'.

\_\_\_\_\_\_\_\_\_\_\_\_\_\_\_\_-

ANSWER Changed accordingly

\_\_\_\_\_\_\_\_\_\_\_\_\_\_\_\_\_\_

Page 1 line 38: 'At the 95% level. . .' It should be 'At the 95% confidence level. . .'

\_\_\_\_\_\_\_\_\_\_\_\_\_\_\_

ANSWER Changed accordingly

\_\_\_\_\_\_\_\_\_\_\_\_\_\_\_\_\_\_

Page 2 line 16: '. . .transporting heat, salt and other substances.' As above – what other substances?

—————————-

ANSWER Changed to" . . . heat and salt."

—————————

Page 2 line 27: '. . .as "overflow" waters.' Why to use the quotation marks here? Overflow water is well-accepted name for this water mass.

——————

ANSWER Quotation marks removed

————————

Page 2 line 29: '. . .entrain on route. . .' Either 'en route' or 'on the way'.

————————-

ANSWER Changed to "en route"

——————

Page 3 lines 4-6: 'The inflowing water from the Atlantic. . .' What about the part of Atlantic water that recirculates along different loops in the Nordic Seas and Arctic Ocean and does not return as 'cold and fresh surface outflow' but rather occupies the subsurface and intermediate layers when flowing to the south?

————————-

ANSWER Rewritten to: The inflowing water from the Atlantic that does not return as overflow mixes with the Pacific inflow and leaves the AM through the Canadian Archipelago and Denmark Strait and the upper western Fram Strait as cold and relatively fresh "surface outflow" (Curry et al., 2014; de Steur et al., 2017).

———————

Page 3 line 22: '. . .expected to be qualitatively different. . .' This statement sounds

a little peculiar. If it was meant that the budget has different components (different flow branches) then it is quantitatively different. On the other hand, the volume (mass) budget should be closed both for the AM and for the Arctic Ocean thus it cannot be 'qualitatively different'. I would suggest reformulating this sentence.

———————-

ANSWER The word qualitative is now deleted and the sentence reformulated

—————————-

Page 4 line 7: 'without any yielding any information. . .' One 'any' too many. . . (without yielding any information).

————————-

ANSWER Changed accordingly

—————————

Page 4 line 9: 'the variability in physical aspects. . .' This sounds somehow cryptic. What are the differences between individual branches that make them difficult to be described in a consistent manner?

————————

ANSWER Deleted statement

—————————

Page 4 line 17: '. . .for historical and logistical reasons. . .' What are 'logistical' reasons? Do you mean the distribution/locations of observations or the structure of paper?

———————

ANSWER The word logistical has been deleted and the sentence reformulated
* * *
Page 4 lines 25-26: 'Over the deepest part. . . . . . towards the Irminger Sea'. This sentence does not belong here as it describes the outflow (DSOWflowing towards the Irminger Sea), not inflow. The same refers to the previous sentence where the surface outflow in the EGC is described. I would suggest keeping the description of inflows and outflows separate.

________________-

ANSWER The discussion of the two outflows have been deleted
* * *
Page 4 line 27 and Fig. 3: Why are these two branches not shown on Figure 3?
* * *
ANSWER Both branches are now shown on the figure
* * *
Page 4 line 34, page 5 line 1: This sentence is difficult to follow (in particular 'are used' at its end), please reformulate.
* * *
ANSWER The sentence has been deleted
* * *
Page 5 lines 25-27: Does the Faroe Current as measured at the section N include the entire flow of AW passing between Iceland and Faroe or is there any part that passes northward beyond the section and is not accounted for?
* * *
ANSWER A sentence has been added to clarify this.
* * *
Page 6 lines 5-6: '. . .a significant fraction originally crossed. . . . . .bifurcated into the FSC. . .' The verb tenses are strange here. I would suggest '. . .a significant fraction that originally crossed the ridge. . . . . .enters the Faroe Current and bifurcates into the FSC. . .'.
* * *
ANSWER Changed accordingly
* * *
Page 8 line 27: '. . . on the Greenland shelf region. . .' Either 'on the Greenland shelf' or 'in the Greenland shelf region. . .'
* * *
ANSWER Changed to: "on the Greenland shelf'
* * *
Page 9 line 19: '. . .uncertainty (estimated from their figures). . .' How was the uncertainty estimated from the figures?
* * *
ANSWER The text "(estimated from their figures)" has been deleted Page 10 line 9: '. . .the "kinematic overflow", has an average volume transport of. . .' What is a difference overflow and 'kinematic overflow'? Is the latter one defined not by density range but some other criteria?

______________-

ANSWER Clarifying text has been added Page 10 line 30: 'The definition of FSCBW is denser than our criterion. . .' Is the assumed source FSCBW denser that the criterion for overflow water or the mixture between FSCBW and AW? The criterion

used by Johnson et al. (2017) is on the other hand less dense therefore 0.3 Sv may by overestimated.

———————————-

ANSWER 'The definition of FSCBW is slightly denser than our criterion for overflow water (27.8 kg m-3) and thus, 0.2 Sv is a lower bound for the volume transport. Previous measurements in the region have suggested transports between 0.1 and 0.3 Sv (Hansen and Østerhus, 2000). We therefore use the timeseries of FSCBW transport based on the method of Sherwin et al. (2008) but attach an uncertainty of $\pm$ 0.1 Sv.'

————————————-

Page 11 line 9 and following: 'Canadian Archipelago. . .' The commonly accepted name is the Canadian Arctic Archipelago (CAA).

———————————

ANSWER Changed to Canadian Arctic Archipelago (CAA)

——————————————

Page 11 lines 16-17: '. . .carries inputs from the integrated CA outflow as well as northward inflow. . .' Perhaps it could be helpful here to mention a different origin (and characteristics) of water masses in the integrated CAA outflow and in the recirculating flow from the West Greenland shelf and slope. A more precise way to describe the outflow from the Davis Strait would be 'the integrated CAA throughflow and modified AW recirculating from the West Greenland Current'.

———————————————————-

ANSWER We have added more text on the origin of water masses

——————————————————-

Page 13 lines 22-23: 'The sum of the transport values. . . . . .did not, however,

differ substantially from the sum based on the full periods' Even if the sum of transport values did not differ substantially, it would be helpful to be able to compare the 6-year averages of volume transport for individual branches with those based on the full periods. Perhaps one more column could be included in Table 1 to show transports averaged for the reference (overlapping) period, especially when taking into account that monthly averages over this period are later employed to analyze the seasonal variations.
* * *
ANSWER Text changed to: The sum of the transport values for all of these branches in these months are all inside the error estimate for the sum based on the full periods (Table 1).

_____________-

Page 14 line 22: '. . .without taking serial correlations into account. . .' Please explain more precisely how would accounting for autocorrelation increase the confidence intervals for calculation of trends in volume transports.
* * *
ANSWER This has been elaborated on

_____________-

Page 17 line 9: '. . . which supports the value of 5 cm as a maximum in the AM as a whole.' Could you elaborate more precisely how is the maximum value of 5 cm for the whole AM obtained from the sea level variations south of 82ÅŮęN.
* * *
ANSWER The sentence has been modified

_________________-

Page 17 line 13: '. . .is not very consistent with this.' I would not call it 'not very consistent' but not consistent at all since there is a differenceon the order of magnitude between the seasonal amplitude estimated from the sea level Variation n and seasonal amplitude based on volume transport measurements.

———————-

ANSWER "not very" has been changed to "not at all"

————————

Page 18 lines 1-8: I would be more careful about downplaying the uncertainties related to different criteria used to distinguishing water masses in inflows and outflow/overflows. The relationship between flow (transport) and hydrographic characteristics at the section is not necessarily linear and it is unclear to me why possible differences should result in systematic biases, not the random errors.

———————-

ANSWER This whole argument has been deleted from the text here and later

—————————-

Page 18 line 8: '. . .budget of the AM and data quality' The phrase 'Data quality' does not reflect the core of the problem as the data quality is the most likely acceptable for this kind of large-scale estimates. The problem is in too sparse measurements, so I would rather suggest '. . .budget of the AM and gaps in the observational coverage'

——————————-

ANSWER Changed accordingly

——————

Page 18 lines 19-20: '. . .in the form of an unknown bias rather than a randomly varying error. . .' As mentioned above, I am not convinced that this is necessarily the case.
* * *
ANSWER As mentioned above, this argument has been deleted from the text
* * *
Page 19 line 5: 'Combining the uncertainties. . . . . .quadratically, as commonly done. . .' 'Combining quadratically' sounds a little peculiar, please reformulate into the assumption about error propagation. Why should it also be 'a conservative estimate' of the overall uncertainty?
* * *
ANSWER The text has been modified to refer to error propagation and the word "conservative" has been removed

______________-

Page 19 lines 18-19: '. . .most of this seasonality would have to come from the DS-outflow, i.e. the estuarine loop. . .' The meaning of the estuarine and thermohaline loops should be introduced before discussing their roles in the seasonal variability.

_______________-

ANSWER Deleted ", i.e., the estuarine loop"
* * *
Page 19 lines 21-22: '. . .if the monitoring of the various import and export branches in the Greenland-Scotland region had been better coordinate with identical monitoring sections for import and export branches.' The meaning of this sentence is entirely incomprehensible to me. Do you mean coordination in time (concurrent monitoring)?
* * *
ANSWER The text has been modified and the meaning hopefully clearer

—————————

Page 19 line 34: Should be 'in some years. . .'

—————————-

ANSWER Changed accordingly

—————————

Page 20 line 3: 'in a mixing rate of 99:1. . .' Where does this estimate of mixing ratio come from (it is not clear from the given salinities for AW and OW)?

—————————

ANSWER It is not clear to us why the referee disagrees. More accurately, the ratio is 0.9887 to 0.0113 ($35.3\times0.9887+0\times0.0113=34.9\times1.0000$), which we round to 0.99:0.01 = 99:1. Retaining more decimals seems to us not justified taking into account the uncertainties in the basic numbers (e.g. salinities). Thus, we have not changed this text, but we have added some more text, which hopefully clarifies the argument.

——————————

Page 20 lines 15-17: Could you provide at least rough estimates for the additional contributions from the entrainment and convection?

—————————-

ANSWER We find it difficult to quantify these contributions without considerable extra text. Instead, we have modified the text slightly and added a reference to a just-published paper emphasizing the point that we wanted to make.

——————————————-

Page 21 lines 10-11: 'Perhaps, slight strengthening of both circulation loops but certainly no weakening'. This sounds as a speculative statement. Please elaborate more precisely and formulate as a full sentence.
* * *
ANSWER The sentence has been deleted

________________-

Page 22 lines 7-8: 'We argue that the exchange branches that have been monitored for a long time most likely do give a good representation of the long-term variations'

The sentence that longer observations provide better estimates of long-term variations is a truism. I would suggest using more precise formulation here.
* * *
ANSWER The sentence has been reformulated to be more precise
* * *
Page 22 lines 30: '. . . a one-time effort with all exchange branches monitored over a year would help substantially.' While this is the most likely true, a concrete argument how would it help would be more convincing (e.g. elucidating relations between transports in different branches, lower uncertainties, etc.)
* * *
ANSWER The sentence has been deleted
* * *
Page 32 Figure 3 and page 33 Figure 4: My suggestion is to slightly enlarge Figure 3 towards the south and incorporate the arrows showing the ES inflow into it. Figure 4 is in my opinion superfluous.
* * *
ANSWER Figure 3 has been enlarged and necessary information from the old Figure 4 has been added to Figure 3. Figure 4 has been deleted and subsequent figures

re-numbered.
* * *
Page 34 Figure 7. Why are the abbreviations of currents' names with dots (periods) on this figure (and in its caption) and without periods on other figures.
* * *
ANSWER The dots on this figure (now Figure 6) and on old Figure 5 (now Figure 4) have been removed and figure captions modified accordingly
* * *
Page 35 Figure 8 and page 36 Figure 10: I would suggest combining panels (a) and (b) into one plot for each of these figures and, in the first place, using one Y-scale for inflows and outflows to be able to compare their variations.
* * *
ANSWER Has been done (now Figures 7 and 9)

---

## Author Comment (AC5) · 11 Mar 2019

The revised "clean" manuscript is Attached

———————————————————

---

## Author Response (AR2)

**Arctic Mediterranean Exchanges: A consistent volume budget and trends in transports from two decades of observations" by Svein Østerhus et al.**
https://doi.org/10.5194/os-2018-114

**Response to Topic Editor**

Please find our answer below ( in yellow answering boxes).

Technical issues:

P2, L2 this enables us (instead of allows us)

| ANSWER | Changed accordingly |
|---|---|

P6, L10 water originally (blank)

| ANSWER | Changed accordingly |
|---|---|

P16, L2 delete: in principle

| ANSWER | Deleted |
|---|---|

P16, L3-4 and is discussed in Sect. 4.1. (instead of: and deserves discussion (Sect. 4.1))

| ANSWER | Changed accordingly |
|---|---|

P16, L17 "is" instead of "was"

| ANSWER | Changed accordingly |
|---|---|

P16, L21-22 (Morison et al., 2007); this is further evidence that an imbalance of 0.1 Sv is unrealistic.

| ANSWER | Changed accordingly |
|---|---|

P19, L7-8 "We should therefore not expect a perfect balance between average AM-import and AM-export." I do not agree with that. With all these caveats, an imbalance is possible but not a priori expected.

| ANSWER | The word "expect" has been replaced by "demand". |
|---|---|

P20, L11 "For other purposes, this might not be the case, however." It is not clear what is meant here. Please rephrase or explain.

| ANSWER | This sentence and the end of the paragraph have been replaced by: "The main motivation for monitoring these flows is, however, to observe their effects on conditions in the AM and on the AMOC. That purpose may be better served by locating some of the monitoring sections somewhat downstream from the intensive mixing areas over the Greenland-Scotland Ridge." |
|---|---|

Repetition:

The Conclusions section contains a lot of repetition from the previous sections, and not that much of new conclusions. Please check whether this can be reduced and changed.

| ANSWER | The Conclusions section has been reduced by more than 40% and is now hopefully less repetitive and more concise. |
|---|---|

**References:**

Several cases: Deep-Sea Research (hyphen)

| **ANSWER** | All "Deep Sea" changed to Deep-Sea |
|---|---|

Andersen and Piccioni should be: Front. Mar. Sci., 3:76, doi: 10.3389/fmars.2016.00076

| **ANSWER** | Changed accordingly |
|---|---|

Change, I. P. o. C.: Climate Change 2013 – The Physical Science Basis … This is not the author; should be Stocker et al. Please correct, also in the text.

| **ANSWER** | Changed to: |
|---|---|
| | Stocker, T. F., Qin, D., Plattner, G.-K., Alexander, L. V., Allen, S. K., Bindoff, N. L., Bréon, F.-M., Church, J. A., Cubasch, U., Emori, S., Forster, P., Friedlingstein, P., Gillett, N., GregoryJ.M. , Hartmann, D. L., Jansen, E., Kirtman, B., Knutti, R., Krishna Kumar, K., Lemke, P., Marotzke, J., Masson-Delmotte, V., Meehl, G. A., Mokhov, I. I., Piao, S., Ramaswamy, V., Randall, D., Rhein, M., Rojas, M., Sabine, C., Shindell, D., Talley, L. D., Vaughan, D. G., and Xie, S.-P.: Climate Change 2013 – The Physical Science Basis: Working Group I Contribution to the Fifth Assessment Report of the Intergovernmental Panel on Climate Change, in, edited by: Stocker, T. F., Qin, D., Plattner, G. K., Tignor, M., Allen, S. K., Boschung, J., Nauels, A., Xia, Y., Bex, V., and Midgley, P. M., Cambridge University Press; Cambridge, United Kingdom and New York, NY, USA, https://doi.org/10.1017/CBO9781107415324.005, 2014. |

Hansen and Meincke: Deep-Sea Research (in full, as in all other references, for consistency)

| **ANSWER** | Change made |
|---|---|

Please check some more at other references.

| **ANSWER** | All references are checked and mistake corrected. |
|---|---|
| | Caltech Library Journal Title Abbreviations is used. |
| | For DOI's we have used: https//doi.org/…… |